# `RADAR`: Defending RAG Dynamically against Retrieval Corruption

**Ziyuan Chen** [1]   **Yueming Lyu** [1]   **Yi Liu** [2]   **Weixiang Han** [1]   **Jing Dong** [3]   **Caifeng Shan** [1]   **Tieniu Tan** [1]

## Abstract

While RAG systems are increasingly deployed in dynamic web search, temporal volatility amplifies their vulnerability to adversarial attacks. Existing static-oriented defenses struggle to handle evolving threats and incur prohibitive storage costs in dynamic settings. We propose `RADAR`, a framework that models reliable context selection as a graph-based energy minimization problem, solved exactly via Max-Flow Min-Cut. By incorporating a Bayesian memory node, `RADAR` recursively updates a belief state instead of archiving raw historical documents, effectively balancing stability against attacks with adaptability to genuine knowledge shifts. Experiments on a novel dynamic dataset show that `RADAR` achieves superior robustness and response quality with minimal storage overhead compared to the baselines. Codes are available at `https://github.com/Ethereallllllll/RADAR_code`.

## 1. Introduction

Large Language Models (LLMs) have demonstrated remarkable capabilities in natural language understanding and generation. However, they remain prone to generating hallucinated or ungrounded content when faced with knowledge-intensive queries. Retrieval-Augmented Generation (RAG) (Lewis et al., 2020b; Guu et al., 2020; Asai et al., 2024) addresses this by incorporating external evidence retrieval into the generation process. Early RAG systems operated in static settings with a fixed corpus as the knowledge base. More recently, research has shifted toward dynamic RAG-based web search (Reddy et al., 2025; Arora et al., 2025; Zhu, 2025), which continuously absorbs updated information from evolving sources such as the web. A prominent application is LLM-augmented search engines: a web search engine retrieves documents relevant to the user's query, and the retrieved content is fed into an LLM to produce a final response grounded in that evidence. Notable examples include Deepseek (DeepSeek-AI, 2024), ChatGPT (OpenAI, 2024), and Grok (xAI, 2025).

Despite its potential, RAG-based web search remains vulnerable to adversarial attacks. Specifically, corpus poisoning (Zou et al., 2025; Hu et al., 2026) and prompt injection (Clop & Teglia, 2024) can manipulate LLMs into generating incorrect or malicious outputs. In dynamic settings, these threats are compounded by temporal volatility, such as mutating or transient content, leading to continuous corruption. This expanded attack surface necessitates defenses that are resilient not only to static attacks but also to evolving adversarial strategies.

Existing defense mechanisms are largely designed for static settings. Heuristic aggregation or filtering (Xiang et al., 2024) often causes utility loss, while optimization-based consistency selection (Shen et al., 2025) typically relies on approximations without strong guarantees. While they alleviate certain vulnerabilities and can be naively extended to dynamic settings, such adaptations are often suboptimal. Without explicit consideration of temporal dynamics, these methods fail to maintain robust performance against continuously mutating threats, resulting in a significant drop in defensive efficacy within dynamic web search contexts.

To address these gaps, we introduce `RADAR`, a robust framework for dynamic RAG. It formulates reliable context selection as a graph-based energy minimization problem, solved exactly via max-flow Min-Cut. `RADAR` utilizes a Bayesian memory node to recursively update a belief state, enabling the system to weigh historical consistency against new evidence. This design effectively resolves the stability-plasticity dilemma, balancing stability against attacks with adaptability to legitimate knowledge shifts.

Our contributions are summarized as follows:

- We propose `RADAR`, which models RAG defense as a Min-Cut problem, delivering exact and efficient inference with superior robustness against corpus-based attacks.
- We design a novel dynamic graph construction augmented with a Bayesian memory node. To the best of our knowl-

---

[1]School of Intelligence Science and Technology, Nanjing University, Suzhou, China [2]City University of Hong Kong, Hong Kong, China [3]Institute of Automation, Chinese Academy of Sciences, Beijing, China. Correspondence to: Yueming Lyu <ymlv@nju.edu.cn>, Yi Liu <97liuyi@ieee.org>, Caifeng Shan <cfshan@nju.edu.cn>.

*Proceedings of the 43rd International Conference on Machine Learning*, Seoul, South Korea. PMLR 306, 2026. Copyright 2026 by the author(s).

edge, this is the first defensive approach explicitly tailored for continuous time-step attacks. By maintaining a recursive belief state, RADAR achieves an effective balance between historical consistency and newly observed evidence.

- We construct a comprehensive dataset for dynamic RAG security, simulating evolving adversarial scenarios. Extensive experiments demonstrate that RADAR achieves superior defense success rates and response quality compared to existing baselines in both static and dynamic scenarios.

## 2. Related Work

### 2.1. Attacks against RAG-based Web Search

Retrieval-Augmented Generation (RAG) systems are vulnerable to adversarial attacks (Bagwe et al., 2025; Chaturvedi et al., 2025; Cho et al., 2024; Jiao et al., 2025; Nazary et al., 2025) that exploit their reliance on external retrieval. We focus on corpus-based attacks, which can be commonly grouped into Prompt Injection Attacks (PIA) and Corpus Poisoning Attacks.

**Prompt Injection Attacks** embed malicious instructions in retrieved documents to override system prompts and hijack outputs. For instance, Backdoored Retrievers (Clop & Teglia, 2024) use implanted backdoors to prioritize injection-carrying passages for link insertion or DoS hijacking. Similarly, Hidden Parrot *et al.* (Prompt Security, 2025) poisons vector stores to covertly steer generation via similarity search.

**Corpus Poisoning Attacks** inject deceptive documents into knowledge bases to manipulate retrieved contexts and downstream outputs. For instance, PoisonedRAG (Zou et al., 2025) optimizes minimal injections for effective knowledge corruption, while BadRAG (Xue et al., 2024) poisons the retrieval process to induce harmful generations. Furthermore, Topic-FlipRAG (Gong et al., 2025) employs two-stage perturbations to reverse topic-specific opinions, and DeRAG (Wang & Yu, 2025) leverages black-box differential evolution to hijack rankings across diverse RAG systems.

### 2.2. Defenses

RAG systems have inspired a range of robustness-enhancing frameworks, which can be broadly categorized into two classes: document filtering prior to generation and defenses against adversarial attacks.

**Document Pre-processing and Filtering** evaluates and refines retrieved content prior to generation. Self-RAG (Asai et al., 2024) employs reflection tokens for adaptive retrieval and self-critique to boost quality; Chain-of-Note (Yu et al., 2024) generates reading notes to filter noise and enhance

robustness. CRAG (Xiang et al., 2024) utilizes lightweight evaluators to trigger corrective actions like web search for reliability, while RA-RAG (Hwang et al., 2025) assesses source credibility and applies weighted majority voting for reliability-aware aggregation.

**Adversarial Defense Strategies** aim to mitigate retrieval corruption and ensure reliability. RobustRAG (Xiang et al., 2024) provides certifiable robustness via formal proofs; InstructRAG (Wei et al., 2025) enables retrieval denoising through self-synthesized rationales. AstuteRAG (Wang et al., 2025) consolidates internal LLM knowledge with retrieved data to resolve conflicts, while ReliabilityRAG (Shen et al., 2025) uses MIS algorithm to filter malicious content with provable guarantees. However, extending these static defenses to dynamic settings often incurs high storage overhead and suboptimal performance.

## 3. Background and Defense Goals

### 3.1. RAG Workflow

**Static Workflow.** A standard RAG system consists of a retriever $\mathcal{R}$ and a generator $\mathcal{G}$ (typically an LLM). Given a user query $q$, the retriever searches a corpus $\mathcal{C}$ to return a set of top-$k$ relevant documents (or passages):

$$\mathcal{D} = \mathcal{R}(q, \mathcal{C}) = \{d_1, d_2, ..., d_k\}. \tag{1}$$

The generator then produces an answer $a$ based on the query and the retrieved context:

$$a = \mathcal{G}(q, \mathcal{D}). \tag{2}$$

The goal is for $\mathcal{D}$ to provide reliable external knowledge that grounds the LLM's response and reduces hallucinations.

**Dynamic Workflow.** In real-world applications, a RAG system typically relies on web search, so its evidence corpus is a continuously evolving set of documents returned by the search engine rather than a fixed local collection. We consider a dynamic setting over discrete time steps $t \in \{0, 1, \ldots, T\}$. At each step, the retriever returns the top-$k$ documents $\mathcal{D}^{(t)} = \mathcal{R}(q)$, and the generator produces an updated answer $a^{(t)} = \mathcal{G}(q, \mathcal{D}^{(t)})$ to reflect the most recent information. The key challenge is to maintain both accuracy and robustness when the reliability of $\mathcal{D}^{(t)}$ varies over time.

### 3.2. Threat Model

**Attacker's Goal.** The attacker aims to induce the LLM to produce incorrect answers that directly contradict the ground truth. For a factual query like "Who is the CEO of Company X?", the attacker seeks to mislead the model into outputting a wrong name, which is an adversarial target.

**Attacker's Background Knowledge.** In practice, attackers can often infer a target RAG system's external knowledge

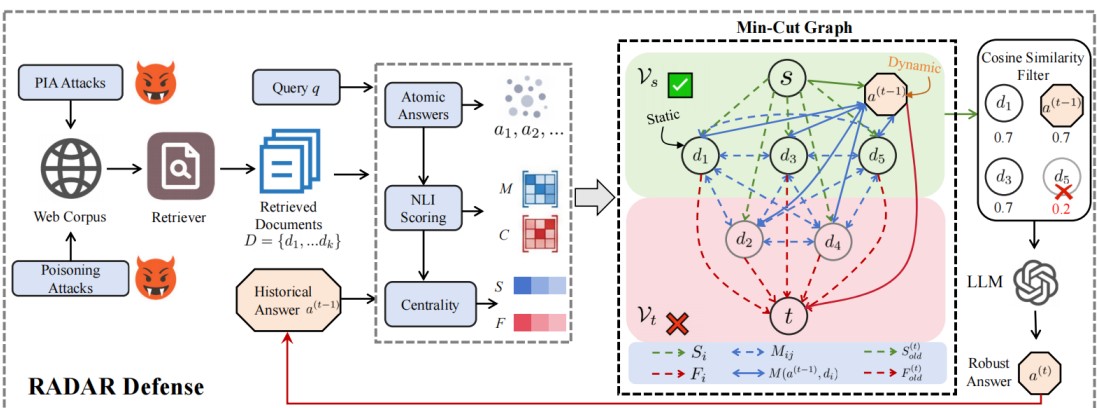

*Figure 1.* Overview of RADAR. It generates an atomic answer for each retrieved document, scores entailment and contradiction using an NLI model, and applies an $s$-$t$ Min-Cut to select a consistent, reliable subset for final answer generation. The dynamic variant augments the graph with a memory node to balance stability and plasticity across time steps.

sources (e.g., Wikipedia) through repeated interactions, enabling them to craft malicious knowledge artifacts. They also understand standard RAG workflows, common retrieval methods and similarity metrics. However, they only have black-box query access: no model parameters, no retriever or generator modification, and no ability to intercept or manipulate user queries.

**Attacker's Capabilities.** The attacker can inject and optimize malicious passages in the external corpus $\mathcal{C}$ to maximize retrieval likelihood, keeping the attack covert at the system-state level: defenders is largely unaware of whether an attack is happening, how many documents are poisoned (denoted by $k'$ or $k_t$), or which retrieved items are poisoned.

**Attack Scenarios.** We categorize attacks into *static* and *dynamic* settings based on whether the adversary can continuously inject malicious documents into the RAG system. In the static setting, the attacker performs a one-time injection into a fixed corpus snapshot, resulting in a constant number of malicious documents in retrieval. In the dynamic setting, the attacker adapts over time, causing the number of malicious retrieved documents $k_t$, and thus the attack scale, to vary across time steps.

### 3.3. Defense Goals

RADAR's main objective is to sanitize the retrieved context $\mathcal{D}$ before feeding it to the generator, thereby ensuring **robustness**: the RAG system should maintain high utility by correctly answering queries based on $D_{clean}$ while effectively neutralizing the influence of $D_{adv}$, achieving both high response accuracy and low Attack Success Rate (ASR).

## 4. Proposed RADAR

**Overview.** To achieve the above goals, we propose RADAR, a robust defense framework tailored for dynamic

RAG (see Figure 1). RADAR first generates an atomic answer for each retrieved document and computes entailment ($M$) and contradiction ($C$) matrices using an NLI model (He et al., 2023). These semantic relations are encoded into an $s$-$t$ graph, and a Min-Cut partitions the document nodes into source and sink sets; the source-side nodes are retained as reliable evidence for answer generation (§4.1–4.2). In dynamic settings, RADAR introduces a *memory node* representing the previous answer, with terminal edges updated via Bayesian inference. This enables the system to adaptively retain or discard historical information in response to new evidence, effectively balancing the stability–plasticity trade-off (§4.3).

### 4.1. Reliable Subset Formulation

Given a user query $q$ and a set of retrieved documents $\mathcal{D} = \{d_1, d_2, \ldots, d_k\}$, our goal is to select a subset of reliable documents $\mathcal{D}_{rel} \subseteq \mathcal{D}$ to generate the final answer. We use binary labeling to annotate whether each document is trustworthy. Let $y = \{y_1, y_2, \ldots, y_k\}$ be a label vector, where $y_i = 1$ indicates that the document $d_i$ is correct and reliable, and $y_i = 0$ indicates it is incorrect and unreliable.

To assign a label $y_i$ to each document $d_i$, we propose to minimize an energy function $E(y)$, formulated as a Markov Random Field (MRF) to find the Maximum A Posteriori (MAP) estimate (Boykov et al., 2002), which balances individual document confidence with pairwise consistency:

$$\min E(y) = \sum_{i=1}^{k} \psi_u(y_i) + \sum_{i,j} \psi_p(y_i, y_j). \qquad (3)$$

Here, **unary potential** $\psi_u(y_i)$ represents the cost of assigning a label $y_i$ to a document, $d_i$ and **pairwise potential** $\psi_p(y_i, y_j)$ represents the penalty for assigning conflicting labels to semantically similar documents. Specifically, $\psi_u(y_i)$

is defined as:

$$\psi_u(y_i) = y_i \cdot F_i + (1 - y_i) \cdot S_i, \tag{4}$$

where $S_i$ represents the benefit of labeling $d_i$ as correct and $F_i$ the benefit of labeling $d_i$ as incorrect. By minimizing this term, the model assigns $y_i = 1$ whenever $S_i > F_i$, indicating that the document's reliability exceeds its risk. For $\psi_p(y_i, y_j)$, it is defined using the consistency score $M_{ij}$:

$$\psi_p(y_i, y_j) = M_{ij} \cdot |y_i - y_j|. \tag{5}$$

This term contributes to the energy cost only when $y_i \neq y_j$, which means one document is labeled correct and the other incorrect, thereby encouraging logical consistency.

Thus, this optimization problem is optimized using the Max-Flow Min-Cut theorem (Elias et al., 1956). The proof is provided in Appendix B. By constructing a flow network, the minimum capacity cut directly corresponds to the minimum energy $E(y^*)$.

## 4.2. Static Defense via Single-Step Min-Cut

**Static Graph Construction.** Before static graph construction, we assess the semantic and logical relationships among retrieved documents. For each document $d_i$, we prompt an LLM to generate an atomic answer $a_i$ and discard $d_i$ if $a_i$ is uninformative. For the remaining documents, we employ an NLI model to compute two matrices: (i) a similarity matrix $M \in [0, 1]^{k \times k}$, where $M_{ij}$ quantifies the degree of logical entailment between $a_i$ and $a_j$; and (ii) a conflict matrix $C \in [0, 1]^{k \times k}$, where $C_{ij}$ measures their logical contradiction. Details of NLI scoring and atomic answer generation are provided in Appendices D and E. To capture global consensus, we compute the eigenvector centrality $v \in \mathbb{R}^k$ of $M$. A higher $v_i$ indicates that $d_i$ is more central to the logical consensus of the retrieved evidence.

Then, we construct a directed graph $G = (V, E)$, where the node set $V$ comprises a source node $s$, a sink node $t$, and nodes representing each retrieved document $\{d_1, \ldots, d_k\}$. The edge set $E$ and their capacities are meticulously designed to encode the energy minimization objective from Eq. (3), ensuring an exact correspondence between the graph cut and the target energy $E(y)$. Thus, the edges and their capacities are defined as follows:

*(i) Source Edges ($s \rightarrow d_i$).* The capacity $S_i$ of a source edge represents the benefit of retaining a document $d_i$ (*i.e.*, labeling it as correct, $y_i = 1$). A higher capacity indicates a stronger pull to include $d_i$ in the reliable set, making it more costly to cut this edge (which corresponds to filtering the document out). We define $S_i$ by integrating the document's eigenvector centrality $v_i$, which reflects its alignment with the global consensus, with a decay function $w_{rank}(i)$ of its original retrieval rank, emphasizing higher-ranked

documents:

$$S_i = v_i \cdot w_{rank}(i), \tag{6}$$

where $w_{rank}(i) = \exp(-\frac{i}{k})$. A large $S_i$ strongly attracts $d_i$ to the source side (the "Correct" partition). The denominator serves as a normalization factor.

*(ii) Sink Edges ($d_i \rightarrow t$).* The capacity $F_i$ of a sink edge represents the benefit of filtering document $d_i$ (*i.e.*, labeling it as incorrect, $y_i = 0$). A higher capacity indicates a stronger push to exclude $d_i$ from the reliable set, making it costly to cut this edge (which corresponds to retaining the document). We define $F_i$ to quantify the conflict between $d_i$ and other highly central, and thus presumably reliable, documents in the retrieved set. It is computed as the average contradiction score $C_{ij}$ between $d_i$ and all other documents $d_j$ ($j \neq i$), weighted by the centrality $v_j$ of $d_j$:

$$F_i = \frac{\sum_{j \neq i} C_{ij} \cdot v_j}{\sum_{j \neq i} v_j}. \tag{7}$$

Intuitively, if $d_i$ conflicts with high-centrality documents, $F_i$ becomes large, strongly pushing $d_i$ towards the sink side (the "Incorrect" partition).

*(iii) Inter-Document Edges ($d_i \leftrightarrow d_j$).* To enforce consistency between semantically related documents, we connect every pair of document nodes $d_i$ and $d_j$ with a pair of anti-parallel edges, each assigned capacity $M_{ij}$. This construction emulates an undirected edge with equivalent cut properties: if $d_i$ and $d_j$ are assigned to different partitions, the cut incurs a penalty of $M_{ij}$, thereby discouraging logical inconsistencies. Here, $M_{ij} \in [0, 1]$ is the symmetric consistency score derived from the entailment between the atomic answers of $d_i$ and $d_j$. Consequently, edges from document nodes to terminals $s$ and $t$ encode the unary potential $\psi_u$, while inter-document edges realize the pairwise potential $\psi_p$.

**Inference via Min-Cut and Answer Generation.** Following the graph construction above, we detail the inference and answer generation pipeline: solving the Min-Cut problem, deriving optimal document labels, and producing the final answer from the selected reliable evidence. Any $s$-$t$ cut partitions the document nodes into a source-side set $\mathcal{V}_s$ and a sink-side set $\mathcal{V}_t$, inducing a natural binary labeling:

$$y_i = \begin{cases} 1, & d_i \in \mathcal{V}_s \text{ (correct)} \\ 0, & d_i \in \mathcal{V}_t \text{ (incorrect)} \end{cases}. \tag{8}$$

The total capacity of the cut can be decomposed into three components:

- $\sum_{d_i \in \mathcal{V}_t} S_i$, corresponding to unary costs $\psi_u(y_i = 0)$;
- $\sum_{d_i \in \mathcal{V}_s} F_i$, corresponding to unary costs $\psi_u(y_i = 1)$;
- $\sum_{d_i \in \mathcal{V}_s, d_j \in \mathcal{V}_t} M_{ij}$, corresponding to pairwise penalties $\psi_p(y_i \neq y_j)$.

Thus, the cut capacity is exactly equal to the energy function:

$$\text{Capacity(Cut)} = E(y). \qquad (9)$$

The set of reliable documents is defined as those assigned to the source partition: $\mathcal{D}_{rel} = \{d_i \mid y_i^* = 1\}$. To ensure semantic consistency, we post-process $\mathcal{D}_{rel}$ by calculating the average pairwise cosine similarity of the selected responses. For each document $d_i \in \mathcal{D}_{rel}$, we compute its average cosine similarity $s_i$ with the other selected documents and exclude those with low agreement, as determined by a hyperparameter $\lambda$. The documents are retained if their average similarity $s_i$ is greater than or equal to $\lambda$:

$$\mathcal{D}'_{rel} = \{d_i \in \mathcal{D}_{rel} \mid s_i \geq \lambda\}. \qquad (10)$$

The details are in Appendix G. The remaining documents in $\mathcal{D}'_{rel}$ are then concatenated with the original query and used to prompt the LLM to generate the final answer.

### 4.3. Dynamic Defense with Memory Node

As defined in Sec. 3.1, real-world RAG systems operate over continuous time steps $t = 1, 2, \ldots, T$, processing an evolving evidence stream $\mathcal{D}^{(t)}$. Naïvely reapplying a static model at each step disregards temporal continuity, often yielding unstable predictions when new evidence is sparse or noisy. Conversely, over-reliance on past answers impedes adaptation to genuine knowledge updates. To resolve this stability–plasticity trade-off, we augment the static graph with a *Memory Node* that encapsulates the system's state from the previous time step, enabling coherent integration of historical knowledge and incoming evidence.

**Dynamic Graph Construction.** At the initial time step $t = 0$, no historical answer exists. In this case, the dynamic mechanism naturally reduces to the static variant. We execute static defense on the initial retrieval set $\mathcal{D}^{(0)}$ to generate the first reliable answer $a^{(0)}$. Then, based on the generated answer and the single-document answers from the current step, we compute the prior probabilities $\pi_S^{(0)}$ and $\pi_F^{(0)}$ for the next round. The priors are obtained by computing the average similarity and conflict:

$$\pi_S^{(0)} = \frac{1}{|\mathcal{D}^{(0)}|} \sum_{d_i \in \mathcal{D}^{(0)}} M(a^{(0)}, d_i), \qquad (11)$$

$$\pi_F^{(0)} = \frac{1}{|\mathcal{D}^{(0)}|} \sum_{d_i \in \mathcal{D}^{(0)}} C(a^{(0)}, d_i). \qquad (12)$$

For $t > 0$, it is necessary to use the dynamic mechanism. The new dynamic graph $G^{(t)} = (V^{(t)}, E^{(t)})$ contains all nodes from the static case—source $s$, sink $t$, and current retrieved documents $\mathcal{D}^{(t)}$, plus the memory node $a^{(t-1)}$. $a^{(t-1)}$ represents the reliable answer generated at time $t-1$.

Thus, the edge capacities are defined as follows, with edges among retrieved new documents, source, and sink following the same definitions as in Sec. 4.2. The key additions are the edges connected to the memory node $a^{(t-1)}$:

*(i) Memory-Source Edges* ($s \rightarrow a^{(t-1)}$). The capacity of this edge $S_{old}^{(t)}$ represents the updated belief that the historical answer $a^{(t-1)}$ remains correct given the new evidence $\mathcal{D}^{(t)}$. We model this using a Bayesian framework since it naturally integrates prior beliefs with new evidence. Let $\pi_S^{(t-1)}$ be the prior probability of correctness, derived from the average consistency of $a^{(t-1)}$ within its previous context at $t - 1$. The likelihood $\mathcal{L}_S^{(t)}$ is the average entailment score between the old answer and the new documents, indicating how well the new evidence supports the history:

$$\mathcal{L}_S^{(t)} = \frac{1}{|\mathcal{D}^{(t)}|} \sum_{d_k \in \mathcal{D}^{(t)}} M(a^{(t-1)}, d_k). \qquad (13)$$

The posterior belief (edge capacity) is computed via Bayes' theorem:

$$S_{old}^{(t)} = \frac{\pi_S^{(t-1)} \cdot \mathcal{L}_S^{(t)}}{\pi_S^{(t-1)} \cdot \mathcal{L}_S^{(t)} + (1 - \pi_S^{(t-1)}) \cdot (1 - \mathcal{L}_S^{(t)})}. \qquad (14)$$

A high capacity attracts the memory node to the source side (the "Correct" partition), signaling that the historical answer is validated by the new information.

*(ii) Memory-Sink Edges* ($a^{(t-1)} \rightarrow t$). The capacity of this edge $F_{old}^{(t)}$ represents the updated probability that the historical answer is incorrect (*i.e.*, should be filtered out). Similarly, we define a prior conflict $\pi_F^{(t-1)}$ and a likelihood of conflict $\mathcal{L}_F^{(t)}$, which is the average contradiction score between the new documents and the old answer:

$$\mathcal{L}_F^{(t)} = \frac{1}{|\mathcal{D}^{(t)}|} \sum_{d_k \in \mathcal{D}^{(t)}} C(a^{(t-1)}, d_k). \qquad (15)$$

The updated capacity $F_{old}^{(t)}$ is:

$$F_{old}^{(t)} = \frac{\pi_F^{(t-1)} \cdot \mathcal{L}_F^{(t)}}{\pi_F^{(t-1)} \cdot \mathcal{L}_F^{(t)} + (1 - \pi_F^{(t-1)}) \cdot (1 - \mathcal{L}_F^{(t)})}. \qquad (16)$$

If new documents explicitly contradict the old answer, this capacity increases, pushing the memory node $a^{(t-1)}$ towards the "Incorrect" partition.

*(iii) Memory-Document Edges* ($a^{(t-1)} \leftrightarrow d_i$). To enforce logical consistency between history and the present, we add undirected edges between the memory node $a^{(t-1)}$ and every current document $d_i \in \mathcal{D}^{(t)}$. The capacity is set to their pairwise consistency $M(a^{(t-1)}, d_i)$. This treats the old answer as a "super-document" that participates in the global consensus. If the old answer is semantically aligned with the majority of valid new documents, these edges reinforce their mutual selection.

**Inference via Min-Cut and Answer Generation.** With the dynamic graph constructed, we solve the Min-Cut problem to obtain the optimal partition $(\mathcal{V}_s^t, \mathcal{V}_t^t)$. The process automatically determines whether the historical information should be retained or discarded. If the memory node $a^{(t-1)}$ remains on the source side $(\mathcal{V}_s)$, the previous conclusion is validated by the new evidence; otherwise, if it is cut to the sink side $(\mathcal{V}_t)$, it indicates concept falsification, and the system removes the historical belief.

Let $\mathcal{A}_{\text{rel}}^{(t)} = \{a_1^{(t)}, \ldots, a_m^{(t)}\}$ denote the set of reliable atomic answers induced by the source partition, *i.e.*, the atomic answers associated with all selected nodes in $\mathcal{V}_s$, including the memory answer $a^{(t-1)}$ if it is retained. Then we post-process $\mathcal{A}_{\text{rel}}^{(t)}$ as in Sec. 4.2 by computing their average pairwise cosine similarity and excluding isolated atomic answers with low similarity to the rest, forming $\mathcal{A}_{\text{rel}}'^{(t)}$. Finally, we prompt the generator $\mathcal{G}$ to strictly synthesize a single coherent conclusion based only on these reliable atomic answers, producing the final answer:

$$a^{(t)} = \mathcal{G}(q, \mathcal{A}_{\text{rel}}'^{(t)}). \tag{17}$$

The coherence of $a^{(t)}$ is then computed and used to update the priors $\pi_S^{(t)}$ and $\pi_F^{(t)}$ for the next time step, with the update formulas being the same as in Eqs. (11) and (12), thereby creating a continuous learning loop. Overall, we present RADAR in Algo. 1.

---

**Algorithm 1** Robust RADAR Defense for Dynamic RAG

1: **Input:** query $q$, retrieval stream $\{\mathcal{D}^{(t)}\}_{t=0}^T$
2: **Output:** answers $\{a^{(t)}\}_{t=0}^T$
3: Initialize $\pi_S \leftarrow 0$, $\pi_F \leftarrow 0$
4: **for** $t = 0$ to $T$ **do**
5:      $\mathcal{A}^{(t)} \leftarrow \text{AtomicGen}(\mathcal{D}^{(t)})$
6:      $(M^{(t)}, C^{(t)}) \leftarrow \text{NLI}(\mathcal{A}^{(t)})$, $v^{(t)} \leftarrow \text{eigcen}(M^{(t)})$
7:      $G^{(t)} \leftarrow \text{BuildGraph}(\mathcal{D}^{(t)}, M^{(t)}, C^{(t)}, v^{(t)})$
8:      **if** $t > 0$ **then**
9:          $G^{(t)} \leftarrow \text{AddHistory}(G^{(t)}, a^{(t-1)}, \mathcal{D}^{(t)}, \pi_S, \pi_F)$
10:          // Inject Bayesian memory node for temporal consistency
11:      **end if**
12:      $(\mathcal{A}_{\text{rel}}^{(t)}, \mathcal{D}_{\text{rel}}^{(t)}) \leftarrow \text{Select}(\text{MinCut}(G^{(t)}))$
13:      $\text{DropIsolated}(\mathcal{A}_{\text{rel}}^{(t)}, \mathcal{D}_{\text{rel}}^{(t)})$
14:      **if** $t = 0$ **then**
15:          $a^{(t)} \leftarrow \mathcal{G}(q, \mathcal{D}_{\text{rel}}^{(t)})$
16:      **else**
17:          $a^{(t)} \leftarrow \mathcal{G}(q, \mathcal{A}_{\text{rel}}^{(t)})$
18:          // Generate from reliable atomic claims
19:      **end if**
20:      $\pi_S \leftarrow \frac{1}{|\mathcal{D}^{(t)}|} \sum_{d_i} M(a^{(t)}, d_i)$
21:      $\pi_F \leftarrow \frac{1}{|\mathcal{D}^{(t)}|} \sum_{d_i} C(a^{(t)}, d_i)$
22:      // Update belief priors for next time step
23:      **Return** $a^{(t)}$
24: **end for**

---

# 5. Experiments

## 5.1. Experimental Setup

**Static Evaluation Datasets.** We evaluate RADAR on four benchmark datasets: RealTimeQA (RQA) (Kasai et al., 2023) for regular real-time QA snapshots; Natural Questions (NQ) (Lee et al., 2019) for answering via full Wikipedia articles; TriviaQA (TQA) (Joshi et al., 2017) for evidence-grounded trivia; and Bio (Lebret et al., 2016) for generating long-form biographies from Wikipedia infoboxes.

**Dynamic Evaluation Datasets.** To evaluate the robustness of RAG systems in dynamic environments, we construct a time-evolving QA benchmark of 500 open-domain questions whose ground-truth answers change over time. For each question and timestamp, we retrieve the top-50 relevant webpages via SerpApi's Google Search API (SerpApi, LLC, 2026), forming temporally indexed evidence snapshots. Using DeepSeek (DeepSeek-AI, 2024), we inject two types of adversarial artifacts into these snapshots: (i) poisoned documents containing fabricated but plausible claims aligned with specific questions, and (ii) prompt-injection payloads embedded in retrieved text that hijack the generator to disregard the user query and output attacker-specified content. Representative examples are provided in Appendix P, and dataset statistics are provided in Appendix O.

**Baselines.** We compare RADAR against a standard Vanilla RAG pipeline, which directly prompts the generator with retrieved documents, and several robustness-oriented baselines: RobustRAG (Xiang et al., 2024), AstuteRAG (Wang et al., 2025), InstructRAG (Wei et al., 2025), and ReliabilityRAG (Shen et al., 2025).

**RAG Settings.** We employ three LLMs as generators in our RAG: DeepSeek (DeepSeek-AI, 2024), GPT-4o (OpenAI, 2024), and Grok-4-fast (xAI, 2025). We use DeBERTa-v3 (He et al., 2023) and NLI model to compute $M$ and $C$. We also conducted experiments under two retrieval settings: top-$k = 10$ and top-$k = 50$ retrieved documents.

**Attack Settings.** We evaluate Prompt Injection Attacks (PIA) and Corpus Poisoning Attacks across different retrieval depths: targeting rank 1 (highest-ranked) and rank 10 (lowest-ranked) for $k = 10$, and ranks 1, 25, and 50 for $k = 50$. Multi-position attacks are further detailed in Appendix L to simulate comprehensive adversarial scenarios.

**Metrics.** For QA datasets, we employ Answer Accuracy (Acc.) to match time-specific ground truth and Attack Success Rate (ASR) to measure the frequency of attacker-targeted outputs. For long-form Bio generation, we use DeepSeek as an LLM-as-a-Judge to score accuracy, rele-

*Table 1.* Performance of RADAR and baseline methods on the RQA dataset using DeepSeek.

| Scenario | Pos | Vanilla RAG | | AstuteRAG | | InstructRAG | | RobustRAG | | ReliabilityRAG | | RADAR (Ours) | |
|---|---|---|---|---|---|---|---|---|---|---|---|---|---|
| | | Acc.↑ | ASR.↓ | Acc.↑ | ASR.↓ | Acc.↑ | ASR.↓ | Acc.↑ | ASR.↓ | Acc.↑ | ASR.↓ | Acc.↑ | ASR.↓ |
| **Top-$k = 10$** | | | | | | | | | | | | | |
| Benign | – | 75.0 | – | 35.0 | – | 73.0 | – | 69.0 | – | 75.0 | – | 75.0 | – |
| PIA | Pos 1 | 25.0 | 74.0 | 25.0 | 1.0 | 23.0 | 68.0 | 64.0 | 7.0 | **69.0** | 15.0 | **69.0** | 11.0 |
| | Pos 10 | 59.0 | 28.0 | 23.0 | 1.0 | 66.0 | 5.0 | 69.0 | 4.0 | 74.0 | 6.0 | **75.0** | 5.0 |
| Poison | Pos 1 | 38.0 | 57.0 | 21.0 | 15.0 | 29.0 | 50.0 | 62.0 | 13.0 | **70.0** | 14.0 | **70.0** | 11.0 |
| | Pos 10 | 58.0 | 35.0 | 36.0 | 4.0 | 54.0 | 14.0 | 70.0 | 5.0 | 75.0 | 6.0 | **75.0** | 6.0 |
| **Top-$k = 50$** | | | | | | | | | | | | | |
| Benign | – | 75.0 | – | 41.0 | – | 62.0 | – | 71.0 | – | 75.0 | – | 75.0 | – |
| PIA | Pos 1 | 35.0 | 65.0 | 21.0 | 5.0 | 33.0 | 46.0 | 69.0 | 9.0 | 67.0 | 18.0 | **72.0** | 5.0 |
| | Pos 25 | 68.0 | 15.0 | 38.0 | 2.0 | 61.0 | 3.0 | 69.0 | 4.0 | 74.0 | 3.0 | **76.0** | 3.0 |
| | Pos 50 | 60.0 | 23.0 | 34.0 | 2.0 | 63.0 | 4.0 | 71.0 | 4.0 | 76.0 | 3.0 | **76.0** | 3.0 |
| Poison | Pos 1 | 44.0 | 53.0 | 21.0 | 10.0 | 39.0 | 35.0 | 67.0 | 20.0 | 64.0 | 19.0 | **71.0** | 7.0 |
| | Pos 25 | 65.0 | 26.0 | 37.0 | 3.0 | 57.0 | 21.0 | 71.0 | 5.0 | 70.0 | 7.0 | **76.0** | 5.0 |
| | Pos 50 | 74.0 | 12.0 | 29.0 | 2.0 | 55.0 | 17.0 | 71.0 | 5.0 | 75.0 | 3.0 | **76.0** | 3.0 |

*Table 2.* Performance of RADAR and baseline methods on Bio dataset using DeepSeek.

| Method | Metric | Benign | | PIA | | | | | Poison | | | | |
|---|---|---|---|---|---|---|---|---|---|---|---|---|---|
| | | | | $k = 10$ | | $k = 50$ | | | $k = 10$ | | $k = 50$ | | |
| | | $k = 10$ | $k = 50$ | Pos 1 | Pos 10 | Pos 1 | Pos 25 | Pos 50 | Pos 1 | Pos 10 | Pos 1 | Pos 25 | Pos 50 |
| Vanilla RAG | Acc.↑ | 77.2 | 78.4 | 24.8 | 9.6 | 19.0 | 21.2 | 10.2 | 59.6 | 31.0 | 57.6 | 48.2 | 29.2 |
| | Rel.↑ | 77.8 | 79.2 | 24.2 | 9.6 | 18.2 | 21.0 | 10.0 | 62.6 | 38.2 | 58.4 | 52.0 | 38.8 |
| | Coh.↑ | 85.0 | 84.4 | 28.4 | 13.8 | 22.6 | 24.8 | 12.4 | 72.4 | 48.0 | 70.4 | 57.4 | 46.2 |
| AstuteRAG | Acc.↑ | 87.8 | 86.0 | 54.8 | 63.8 | 67.6 | 68.2 | 73.8 | 66.0 | 68.6 | 65.0 | 70.0 | 65.4 |
| | Rel.↑ | 88.2 | 85.2 | 57.8 | 65.2 | 73.0 | **77.2** | **77.8** | 67.6 | 69.8 | 63.0 | 68.2 | 65.0 |
| | Coh.↑ | 90.8 | 88.0 | 63.4 | 70.2 | 77.8 | 82.8 | 82.6 | 74.6 | 75.6 | 73.2 | 79.6 | 76.6 |
| InstructRAG | Acc.↑ | 74.2 | 73.8 | 68.6 | 71.8 | 74.0 | **76.6** | 76.4 | 74.4 | 73.2 | 77.4 | 76.6 | **80.2** |
| | Rel.↑ | 69.4 | 70.4 | 64.6 | 64.6 | 70.0 | 73.2 | 77.4 | 72.2 | 70.4 | 74.0 | 74.0 | 77.8 |
| | Coh.↑ | 78.4 | 78.2 | 73.0 | 75.6 | 78.8 | 81.4 | 82.4 | 80.4 | 78.0 | 80.8 | 81.4 | 84.2 |
| RobustRAG | Acc.↑ | 60.4 | 60.6 | 57.6 | 55.8 | 57.2 | 61.2 | 56.0 | 60.4 | 52.8 | 52.2 | 65.4 | 68.0 |
| | Rel.↑ | 51.4 | 62.2 | 49.2 | 47.6 | 57.0 | 62.0 | 57.6 | 53.8 | 47.4 | 55.2 | 67.2 | 68.0 |
| | Coh.↑ | 72.4 | 71.6 | 70.2 | 66.2 | 69.0 | 73.0 | 68.0 | 73.4 | 66.6 | 67.6 | 77.0 | 80.4 |
| ReliabilityRAG | Acc.↑ | 70.6 | 75.4 | 66.6 | 67.0 | 71.8 | 74.4 | 74.6 | 65.2 | 75.0 | 68.4 | 80.0 | 73.0 |
| | Rel.↑ | 70.6 | 78.4 | 67.6 | 68.0 | 71.2 | 76.0 | 76.4 | 67.0 | 76.8 | 71.4 | **81.8** | 74.6 |
| | Coh.↑ | 78.4 | 83.2 | 75.2 | 74.0 | 77.4 | 83.0 | 81.4 | 75.8 | 84.0 | 77.8 | **87.4** | 79.6 |
| RADAR (Ours) | Acc.↑ | 76.6 | 76.8 | **72.0** | 75.6 | **76.8** | 75.8 | 80.6 | 84.4 | 79.0 | 81.2 | 80.4 | 78.6 |
| | Rel.↑ | 74.0 | 77.0 | **71.0** | **76.4** | 75.8 | 75.8 | 77.0 | 82.6 | **77.2** | 80.6 | 81.2 | **79.4** |
| | Coh.↑ | 81.4 | 83.4 | **79.2** | **83.0** | **83.4** | **83.4** | **84.2** | **88.2** | **84.4** | 85.4 | 85.6 | **85.6** |

vance, and coherence based on Wikipedia references.

## 5.2. Defense Performance in Static Environments

Our static experiments highlight two RADAR strengths: (i) preserving utility in benign settings and (ii) enhancing robustness against prompt injection and corpus poisoning without compromising quality.

**Benign Performance.** RADAR maintains competitive accuracy, matching Vanilla RAG on RQA (75.0%) as shown in Table 1. In long-form Bio tasks, while AstuteRAG leads due to its specialized noise mitigation, RADAR consistently exceeds defense-oriented baselines RobustRAG and ReliabilityRAG in accuracy, relevance, and coherence. Our sanitization mechanism thus avoids the utility loss typical of heuristic filtering.

**Robustness under Poisoning Attacks.** RADAR yields the best robustness-utility trade-off across datasets and attack positions, achieving low ASR while maintaining the highest accuracy. It remains effective even when Pos 1 is compromised. As shown in Table 1, for top-$k = 10$ on RQA with PIA at Pos 10, it achieves 75.0% accuracy with 5.0% ASR. And for top-$k = 50$ on RQA with PIA at Pos 1, it maintains 72.0% accuracy and 5.0% ASR despite increased

*Table 3.* Performance under evolving evidence streams with top-$k = 50$ using DeepSeek under the cumulative snapshot setting.

| Attack | Pos | Vanilla RAG | | AstuteRAG | | InstructRAG | | RobustRAG | | ReliabilityRAG | | RADAR (Ours) | |
|---|---|---|---|---|---|---|---|---|---|---|---|---|---|
| | | Acc.↑ | ASR.↓ | Acc.↑ | ASR.↓ | Acc.↑ | ASR.↓ | Acc.↑ | ASR.↓ | Acc.↑ | ASR.↓ | Acc.↑ | ASR.↓ |
| Benign | – | 70.76 | – | 73.48 | – | 73.41 | – | 67.50 | – | 70.63 | – | **74.02** | – |
| PIA | Pos 1 | 12.41 | 87.01 | 54.25 | 12.99 | 61.42 | 34.46 | 61.29 | 18.62 | 53.61 | 42.67 | **63.60** | 17.85 |
| | Pos 25 | 29.88 | 67.37 | 62.51 | 9.40 | 65.77 | 28.79 | 66.92 | 9.98 | 67.62 | 9.34 | **70.12** | 8.94 |
| | Pos 50 | 19.32 | 79.40 | 59.18 | 9.66 | 59.69 | 34.68 | 67.43 | 7.87 | 68.71 | 5.95 | **70.05** | 6.01 |
| Poison | Pos 1 | 25.72 | 53.17 | 47.92 | 16.63 | 53.49 | 24.50 | 44.98 | 34.29 | 53.87 | 27.51 | **63.60** | 17.53 |
| | Pos 25 | 35.44 | 43.44 | 58.41 | 10.62 | 55.34 | 25.46 | 66.28 | 11.00 | 67.69 | 7.55 | **69.41** | 7.22 |
| | Pos 50 | 30.77 | 46.51 | 57.77 | 9.60 | 55.92 | 23.74 | 66.92 | 8.64 | 67.88 | 5.25 | **70.37** | 5.95 |

*Table 4.* Performance under evolving evidence streams with top-$k = 50$ using DeepSeek under the lightweight history setting.

| Attack | Pos | Vanilla RAG | | AstuteRAG | | InstructRAG | | RobustRAG | | ReliabilityRAG | | RADAR (Ours) | |
|---|---|---|---|---|---|---|---|---|---|---|---|---|---|
| | | Acc.↑ | ASR.↓ | Acc.↑ | ASR.↓ | Acc.↑ | ASR.↓ | Acc.↑ | ASR.↓ | Acc.↑ | ASR.↓ | Acc.↑ | ASR.↓ |
| Benign | – | 70.70 | – | 64.17 | – | **83.10** | – | 59.88 | – | 72.55 | – | 74.02 | – |
| PIA | Pos 1 | 37.68 | 60.72 | 60.08 | 7.74 | 57.38 | 32.69 | 53.49 | 20.15 | 50.74 | 43.12 | **63.60** | 17.85 |
| | Pos 25 | 36.98 | 58.35 | 64.68 | 8.25 | 55.34 | 25.46 | 59.31 | 12.92 | 66.53 | 10.81 | **70.12** | 8.94 |
| | Pos 50 | 14.84 | 82.47 | 62.06 | 9.85 | 42.22 | 49.07 | 59.12 | 12.80 | 68.45 | 6.78 | **70.05** | 6.01 |

noise and attack surface. Due to space constraints, the test results for NQ and TQA and the test results for GPT-4o and Grok-4-fast are presented in Appendix K.

**Long-form Generation Robustness under Poisoning Attacks.** As shown in Table 2, RADAR dominates Bio task performance under PIA and poisoning across most positions. Notably, under PIA attacks at position 50 when top-$k = 50$, RADAR maintains a high factual accuracy of 80.6%, whereas Vanilla RAG's performance collapses to 10.2%. Furthermore, in poisoning scenarios when top-$k = 10$ at Pos 1, RADAR achieves a peak accuracy of 84.4%, outperforming all baseline models. It consistently improves factual accuracy, relevance, and coherence. The method effectively blocks adversarial inputs while maintaining high-quality discourse in long-form outputs.

### 5.3. Defense Performance in Dynamic Environments

**Experimental Protocol.** Most existing RAG defenses are designed for static snapshots of retrieved evidence and do not naturally handle time-evolving streams. To enable a fair comparison in dynamic scenarios, we explore two ways to adapt static baselines to the temporal setting: (1) cumulative snapshot: at each time step $t$, newly retrieved documents are prepended to all previously seen documents $\{D_0, D_1, \ldots, D_{t-1}\}$, forming an expanding evidence pool $D_{\leq t}$; (2) lightweight history: instead of storing the full document history, only the answer from the previous time step is appended to the current prompt. Both approaches are applied to all baselines.

**Benign Performance.** As shown in Table 3 and Table 4, in

the no-attack setting, RADAR achieves a peak accuracy of 74.02%, surpassing vanilla RAG in both temporal settings. This is because the cumulative snapshot approach may feed obsolete or incorrect evidence from earlier time steps into the LLM, while the lightweight history approach may mislead the model with previous answers that are no longer valid. These results show that RADAR not only defends against attacks but also improves correctness in dynamic RAG.

**Performance Under Time-Evolving Attacks.** Table 3 presents a comparison between RADAR and other baselines under the cumulative snapshot setting, reporting accuracy and ASR on evolving evidence streams with time-varying attacks. RADAR offers a better robustness–utility trade-off than RobustRAG and ReliabilityRAG, achieving higher accuracy and maintaining lower or stable ASR across injection positions. Notably, under the most challenging Pos 1 PIA attack, RADAR achieves 63.60% accuracy, significantly outperforming RobustRAG (61.29%) and ReliabilityRAG (53.61%). RADAR excels when adversarial content is injected into high-ranked evidence, better protecting critical passages. For mid-rank and tail-rank injections, RADAR sustains about 70% accuracy with low ASR. Table 4 further compares RADAR with baselines under the lightweight history setting, where only the previous answer is appended to the current prompt. Under PIA attacks, RADAR consistently achieves the highest accuracy across injection positions, surpassing the best baseline by 3.52% at Pos 1 and 5.44% at Pos 25, while maintaining low ASR (17.85% and 8.94%, respectively), demonstrating its superior robustness against adversarial injections even under lightweight history adaptation.

*Table 5.* RADAR's performance on different NLI models under PIA attack on RQA using Deepseek.

| NLI | Top-$k$ = 10 | | | | Top-$k$ = 50 | | | | | |
| --- | --- | --- | --- | --- | --- | --- | --- | --- | --- | --- |
| | Pos 1 | | Pos 10 | | Pos 1 | | Pos 25 | | Pos 50 | |
| | Acc.↑ | ASR.↓ | Acc.↑ | ASR.↓ | Acc.↑ | ASR.↓ | Acc.↑ | ASR.↓ | Acc.↑ | ASR.↓ |
| DeBERTa-v3 | 69.0 | 11.0 | 75.0 | 5.0 | 72.0 | 5.0 | 76.0 | 3.0 | 76.0 | 3.0 |
| BART | 69.0 | 11.0 | 75.0 | 6.0 | 72.0 | 7.0 | 76.0 | 4.0 | 76.0 | 4.0 |
| ModernBERT | 69.0 | 12.0 | 75.0 | 6.0 | 72.0 | 7.0 | 76.0 | 4.0 | 76.0 | 4.0 |

*Figure 2.* Sensitivity of the post-processing threshold $\lambda$.

*Table 6.* RADAR's performance under random perturbations of NLI with Top-$k$ = 10 under PIA attack on RQA using Deepseek.

| Perturbation Rate | Pos 1 | | Pos 10 | |
| --- | --- | --- | --- | --- |
| | Acc.↑ | ASR.↓ | Acc.↑ | ASR.↓ |
| 0 | 69.0 | 11.0 | 75.0 | 5.0 |
| 0.1 | 67.0 | 10.0 | 74.0 | 6.0 |
| 0.3 | 68.0 | 12.0 | 73.0 | 6.0 |
| 0.5 | 64.0 | 18.0 | 78.0 | 7.0 |

### 5.4. Hyperparameter Sensitivity

We evaluate the sensitivity of the post-processing threshold $\lambda$ on RQA under prompt injection attacks using DeepSeek. As shown in Figure 2, varying $\lambda$ from 0.1 to 0.5 has minimal impact on accuracy and ASR, with performance remaining stable across injection positions. ASR is slightly more sensitive, following a decrease-then-degrade pattern as $\lambda$ increases. Across retrieval sizes and injection ranks, $\lambda = 0.3$ provides the most consistent robustness gains with negligible accuracy loss, so we adopt $\lambda = 0.3$ by default.

### 5.5. NLI Sensitivity

We ablate different NLI models (DeBERTa-v3 (He et al., 2023), BART (Lewis et al., 2020a), and ModernBERT (Warner et al., 2025)) under PIA on RealTimeQA in Table 5 and observe only minor changes in Acc. and ASR, indicating low sensitivity to the NLI choice. We further perturb NLI outputs by randomly replacing them with probabilities of 0.1, 0.3, and 0.5 in Table 6, and observe only slight accuracy drops, reinforcing this low sensitivity.

*Table 7.* RADAR's performance under none-adaptive attack and adaptive attack on RQA using Deepseek.

| Top-$k$ | Position | None-Adaptive | | Adaptive | |
| --- | --- | --- | --- | --- | --- |
| | | Acc.↑ | ASR.↓ | Acc.↑ | ASR.↓ |
| 10 | Pos 1 | 69.0 | 11.0 | 69.0 | 10.0 |
| | Pos 10 | 75.0 | 5.0 | 74.0 | 6.0 |
| 50 | Pos 1 | 72.0 | 5.0 | 72.0 | 8.0 |
| | Pos 25 | 76.0 | 3.0 | 75.0 | 5.0 |
| | Pos 50 | 76.0 | 3.0 | 74.0 | 5.0 |

### 5.6. NLI Robustness under Adaptive Attack

To directly assess whether the NLI-based entailment/contradiction signals remain reliable under adversarially written or stylistically camouflaged injected text, we additionally evaluate RADAR under the adaptive attack setting proposed by ReliabilityRAG. Specifically, besides the standard non-adaptive injection, we consider the adaptive attack that induces ambiguous answers (e.g., "A or B" while the correct answer is A) to bypass NLI contradiction checks.

Our results in Table 7 show that RADAR remains largely stable under this stronger attack: across both top-$k = 10$ and top-$k = 50$ settings, the performance under adaptive attack is very close to that under non-adaptive attack, with only marginal differences. This suggests that RADAR remains effective even when the injected text is adversarially crafted to camouflage itself against NLI-based defenses.

## 6. Conclusion

In this paper, we present RADAR, a defense framework for dynamic RAG that treats context sanitization as a Min-Cut problem. By merging graph-theoretic inference with a Bayesian Memory Node, RADAR balances adversarial resilience and knowledge adaptation with minimal storage overhead. We also provide a comprehensive dynamic attack dataset as a benchmark. As AI systems pivot toward real-time data, such mathematically grounded, time-aware resilience is vital for next-generation trustworthy AI agents.

## Impact Statement

This work introduces RADAR, a robust and storage-efficient defense framework for dynamic Retrieval-Augmented Generation (RAG) systems operating in adversarial, time-evolving environments such as web search. By formulating context sanitization as a graph-based energy minimization problem solved via Min-Cut and incorporating a Bayesian memory node, RADAR effectively balances stability against retrieval corruption with adaptability to genuine knowledge updates, without archiving historical documents. This approach significantly reduces storage overhead (from tens of MBs to ~1 KB per query) while improving robustness and answer accuracy under evolving attacks. As real-time RAG becomes integral to AI assistants, search engines, and decision-support tools, RADAR provides a practical, mathematically grounded mechanism to enhance trustworthiness, mitigate prompt injection and corpus poisoning risks, and promote the safe deployment of LLM-powered systems in dynamic real-world settings.

## Acknowledgements

This work was supported by the New Generation Artificial Intelligence-National Science and Technology Major Project (2025ZD0123504), the National Natural Science Foundation of China (Grants 62502200), the Jiangsu Provincial Science and Technology Major Project (Grant BG2024042), and the Natural Science Foundation of Jiangsu Province (Grants BK20251203).

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

## A. Overview

This appendix provides supplementary technical details, derivations, and experimental results supporting the RADAR method presented in the main paper. The sections are organized as follows:

Appendix B derives the submodularity of the proposed energy function and explains its exact minimization via $s - t$ Min-Cut construction.

Appendix C justifies the Bayesian update rules used for dynamic capacity adjustment of historical answers.

Appendix D-H detail key implementation components:

- Symmetric NLI scoring for consistency $M$ and conflict $C$ matrices

- Document-wise atomic answer generation

- Eigenvector centrality computation for consensus scoring

- Post-processing for semantic outlier removal

- Constrained LLM synthesis prompt for final answer generation

Appendix I-J analyze computational aspects:

- Min-cut complexity using HLPP

- Runtime and efficiency

Appendix K-M add more experimental results:

- Extended static PIA & Poison attack results on NQ, TQA, Bio using DeepSeek, GPT-4o, and Grok-4-fast

- Multi-position injection attack results

- Extended dynamic PIA attack results using GPT-4oand Grok-4-fast

Appendix N shows the failure cases of our method.

Appendix O-P gives details of our dynamic dataset:

- Dataset statistics

- Examples of the dataset

## B. Min-Cut Solvability of the Energy

Under the Markov Random Field (MRF) framework, minimizing the energy function $E(y)$ corresponds to finding the Maximum A Posteriori (MAP) estimate of the document labels. Specifically, we seek a binary labeling $y \in \{0, 1\}^k$ by minimizing:

$$E(y) = \sum_{i=1}^{k} \left( y_i F_i + (1 - y_i) S_i \right) + \sum_{1 \le i < j \le k} M_{ij} |y_i - y_j|. \tag{18}$$

The pairwise term is a weighted Potts model, which is graph-representable when it is submodular for each pair $(i, j)$. Concretely, define

$$E_{ij}(y_i, y_j) = M_{ij} |y_i - y_j|. \tag{19}$$

When $M_{ij} \ge 0$, we have

$$E_{ij}(0,0) = 0, \quad E_{ij}(1,1) = 0,$$
$$E_{ij}(0,1) = M_{ij}, \quad E_{ij}(1,0) = M_{ij}. \tag{20}$$

which satisfies the submodularity inequality

$$E_{ij}(0,0) + E_{ij}(1,1) \leq E_{ij}(0,1) + E_{ij}(1,0), \tag{21}$$

since $0 + 0 \leq M_{ij} + M_{ij}$ holds whenever $M_{ij} \geq 0$. Therefore, the overall energy $E(y)$ belongs to the class of submodular binary energies and can be minimized exactly via an $s$–$t$ Min-Cut (Kolmogorov & Zabin, 2004). Additionally, under the standard $s$-$t$ graph construction, the unary term $y_i F_i + (1 - y_i) S_i$ is represented by terminal edges $(s \to d_i)$ and $(d_i \to t)$ with capacities $S_i$ and $F_i$, respectively, while the pairwise term $M_{ij}|y_i - y_j|$ is represented by an undirected edge between $d_i$ and $d_j$ with capacity $M_{ij}$. Consequently, the cut cost equals $E(y)$, and the Min-Cut yields the global minimizer $y^*$.

## C. Justification of Bayesian Memory Update

In dynamic defense, we update the capacity of the memory node based on new evidence. Here, we interpret the capacity $S_{old}^{(t)}$ as the posterior probability that the historical answer $a^{(t-1)}$ remains correct. We define the binary random variable $H \in \{0, 1\}$, where $H = 1$ denotes the hypothesis that $a^{(t-1)}$ is correct.

**Prior.** The prior probability $P(H = 1)$ is given by $\pi_S^{(t-1)}$, which is derived from the coherence of the previous generation step.

**Likelihood.** Let $E$ be the event of observing the semantic relationship between the old answer and the current retrieved documents $\mathcal{D}^{(t)}$. We use the aggregated entailment score $\mathcal{L}_S^{(t)}$ as the likelihood of observing such support given that the history is correct:

$$P(E \mid H = 1) = \mathcal{L}_S^{(t)}. \tag{22}$$

To make the update tractable, we adopt a *Symmetric Likelihood Assumption*. We assume that if the historical answer were incorrect ($H = 0$), the probability of observing high entailment from valid new documents would be the complement of the support score:

$$P(E \mid H = 0) = 1 - \mathcal{L}_S^{(t)}. \tag{23}$$

This assumption reflects the intuition that an incorrect answer will contradict or fail to entail the true information present in $\mathcal{D}^{(t)}$.

**Posterior.** By applying Bayes' theorem, the posterior probability $P(H = 1 \mid E)$ is:

$$\begin{aligned} P(H = 1 \mid E) &= \frac{P(E \mid H = 1) \cdot P(H = 1)}{P(E)} \\ &= \frac{P(E \mid H = 1) \cdot P(H = 1)}{P(E \mid H = 1)P(H = 1) + P(E \mid H = 0)P(H = 0)}. \end{aligned} \tag{24}$$

Substituting the prior and likelihood terms:

$$S_{old}^{(t)} = \frac{\mathcal{L}_S^{(t)} \cdot \pi_S^{(t-1)}}{\mathcal{L}_S^{(t)} \cdot \pi_S^{(t-1)} + (1 - \mathcal{L}_S^{(t)}) \cdot (1 - \pi_S^{(t-1)})}. \tag{25}$$

This recovers the update formula in Eq. 14. The update for the conflict capacity $F_{old}^{(t)}$ in Eq. 16 follows an identical derivation by defining $H = 1$ as the hypothesis that the answer is *incorrect* and using the conflict matrix for likelihood estimation.

## D. NLI Scoring for $M$ and $C$

We use a Natural Language Inference (NLI) model to quantify the logical relation between two atomic answers. Given an ordered pair (premise, hypothesis), the NLI model outputs a probability distribution over entailment, contradiction, and neutral. We take the entailment probability as the consistency strength and the contradiction probability as the conflict strength. For any pair $(a_i, a_j)$, we denote:

$$M_{i \to j} \in [0, 1] \quad \text{as the entailment strength from } a_i \text{ to } a_j, \tag{26}$$

$$C_{i \to j} \in [0, 1] \quad \text{as the contradiction strength from } a_i \text{ to } a_j. \tag{27}$$

In general, the scores are not symmetric:

$$M_{i \to j} \neq M_{j \to i}, \qquad C_{i \to j} \neq C_{j \to i}. \tag{28}$$

However, our graph construction uses undirected document, which requires a symmetric edge weight. Moreover, the contradiction scores used in risk aggregation should not be dominated by a single directional prediction.

To remove directional bias, we symmetrize the two directions by the geometric mean. For each pair $(i, j)$, we define the symmetric scores:

$$M_{ij} \triangleq \sqrt{M_{i \to j} \, M_{j \to i}}, \tag{29}$$

$$C_{ij} \triangleq \sqrt{C_{i \to j} \, C_{j \to i}}. \tag{30}$$

By construction, these satisfy $M_{ij} = M_{ji}$ and $C_{ij} = C_{ji}$.

The geometric mean enforces a conservative agreement-in-both-directions criterion: the symmetric score is large only when both directions are high, and it is strongly down-weighted if either direction is low. This mitigates directional artifacts while keeping the scores in $[0, 1]$ for direct use as graph capacities.

## E. Atomic Answer Generation

Given the retrieved set $\mathcal{D} = \{d_1, \ldots, d_k\}$ from the standard RAG workflow, we further decompose the generation step into a set of document-wise responses, referred to as atomic answers. Specifically, for each retrieved document $d_i$, we query the generator $\mathcal{G}$ with the original user query $q$ and *only* the single-document context $d_i$:

$$a_i = \mathcal{G}(q, d_i), \quad i = 1, \ldots, k. \tag{31}$$

Here $a_i$ is intended to capture the minimal claim(s) about $q$ that can be supported by $d_i$ alone, decoupling the influence of other retrieved documents. In implementation, we prompt the LLM to (i) answer $q$ using only the evidence in $d_i$, (ii) avoid introducing external knowledge, and (iii) return a concise, self-contained statement.

The resulting atomic answers $\{a_i\}_{i=1}^{k}$ serve as standardized semantic units for subsequent reasoning. We discard documents whose atomic answers are uninformative. For the remaining set, we compute document-level entailment and contradiction relations by applying an NLI model to pairs of atomic answers, which are then used to construct the similarity matrix $M$ and conflict matrix $C$ described in Sec. 4.2.

## F. Eigenvector Centrality Computation

Given the similarity matrix $M \in \mathbb{R}^{k \times k}$, where $M_{ij} \in [0, 1]$ measures the semantic and logical agreement between $a_i$ and $a_j$, we compute a global consensus score for each document node using eigenvector centrality. Intuitively, a node is considered central if it is similar to other central nodes.

To improve numerical stability, we add a small self-loop to each node and define the weighted adjacency matrix

$$A = M + \epsilon I, \tag{32}$$

where $I$ is the identity matrix and $\epsilon > 0$ is a small constant.

Eigenvector centrality is defined as the principal eigenvector of $A$, i.e., the nonzero vector $v \in \mathbb{R}^k$ satisfying

$$Av = \lambda_{\max} v, \tag{33}$$

where $\lambda_{\max}$ is the largest eigenvalue of $A$. We approximate $v$ using $T$ steps of power iteration with $\ell_2$ normalization. Starting from the uniform initialization

$$v^{(0)} = \frac{1}{k} \mathbf{1}, \tag{34}$$

we update

$$\tilde{v}^{(\tau+1)} = Av^{(\tau)}, \qquad v^{(\tau+1)} = \frac{\tilde{v}^{(\tau+1)}}{\left\| \tilde{v}^{(\tau+1)} \right\|_2 + \delta}, \quad \tau = 0, 1, \ldots, T-1, \tag{35}$$

where $\delta > 0$ is a small constant to avoid division by zero. After $T$ iterations, we take $v = v^{(T)}$ as the estimated centrality vector.

Finally, we rescale $v$ into $[0, 1]$ for downstream use:

$$\text{centrality}_i = \frac{v_i - \min_j v_j}{\max_j v_j - \min_j v_j + \delta}, \qquad i = 1, \dots, k. \tag{36}$$

In our implementation, we set $\epsilon = 0.01$, the number of power-iteration steps to $T = 10$, and use $\delta = 10^{-8}$ as a numerical stability constant.

---

**Algorithm 2** Eigenvector Centrality via Power Iteration

---

**Require:** Similarity matrix $M \in \mathbb{R}^{k \times k}$; iterations $T$; constants $\epsilon > 0, \delta > 0$
**Ensure:** Normalized centrality scores $\text{centrality} \in [0, 1]^k$
1: $A \leftarrow M + \epsilon I$
2: $v \leftarrow \frac{1}{k}\mathbf{1}$
3: **for** $\tau \leftarrow 1$ to $T$ **do**
4:    $v \leftarrow Av$
5:    $v \leftarrow \dfrac{v}{\|v\|_2 + \delta}$
6: **end for**
7: $\text{centrality} \leftarrow \dfrac{v - \min(v)}{\max(v) - \min(v) + \delta}$
8: **Return** centrality

---

## G. Post-processing for Semantic Consistency

After Min-Cut inference, we obtain the reliable set

$$\mathcal{D}_{rel} = \{d_i \mid y_i^* = 1\}, \tag{37}$$

together with their corresponding atomic answers $\{a_i\}_{d_i \in \mathcal{D}_{rel}}$. To further ensure semantic consistency among the selected evidence, we apply a post-processing step that removes isolated items based on embedding cosine similarity.

Let $\mathbf{e}_i = \text{Enc}(a_i)$ denote the embedding of atomic answer $a_i$. For any pair of selected answers, we compute the cosine similarity

$$\text{sim}(i, j) = \cos(\mathbf{e}_i, \mathbf{e}_j), \quad d_i, d_j \in \mathcal{D}_{rel}. \tag{38}$$

For each selected document $d_i$, we measure its average semantic agreement with the remaining selected set:

$$s_i = \frac{1}{|\mathcal{D}_{rel}| - 1} \sum_{\substack{d_j \in \mathcal{D}_{rel} \\ j \neq i}} \text{sim}(i, j), \quad d_i \in \mathcal{D}_{rel}. \tag{39}$$

Documents whose atomic answers exhibit low agreement with the rest are treated as isolated outliers and excluded:

$$\mathcal{D}'_{rel} = \{d_i \in \mathcal{D}_{rel} \mid s_i \geq \lambda\}. \tag{40}$$

In our implementation, we set $\lambda = 0.3$. The remaining atomic answers associated with $\mathcal{D}'_{rel}$ are then concatenated with the original query to prompt the LLM for the final answer.

## H. LLM-based Synthesis from Reliable Atomic Answers in dynamic defense

After Min-Cut inference, we obtain a set of reliable atomic answers $\mathcal{A}_{\text{rel}}^{(t)} = \{a_1^{(t)}, \dots, a_m^{(t)}\}$. We generate the final response via a constrained LLM synthesis step, whose goal is to extract the most consistent and dominant conclusion supported by these atomic answers.

To reduce uncontrolled speculation, the generator is explicitly instructed to only use the provided atomic answers as references, and to not add, correct, question, or challenge any information contained in them even if it appears outdated. This turns the synthesis step into a purely aggregative operation over vetted evidence.

We concatenate all reliable atomic answers into a single context string `context_str` and use the following prompt:

> **Question:** {question}
>
> **The following are all the reliable reference atomic answers** (synthesize **strictly based on this content only**. Do NOT add, correct, question, or challenge any information in it, even if you believe it may be outdated): {context_str}
>
> Strictly follow the reference answers provided above and synthesize the most consistent and main conclusion as the final answer.
>
> Output **only the final answer itself**. Do NOT write any explanations, reminders, supplements, or comments about dates.

The model must output a single final answer string without rationale, meta-commentary, or auxiliary notes. This ensures that the final response is a direct synthesis of the selected reliable atomic answers.

## I. Computational Complexity of Min-Cut

By the Max-Flow Min-Cut theorem, the capacity of a $s$-$t$ Min-Cut equals the value of a maximum $s$–$t$ flow; hence we recover the optimal cut and thus $y^*$ by computing a max flow and reading off the $s$-reachable set in the residual graph. RADAR contains $k$ document nodes and two terminals, hence

$$n = k + 2. \tag{41}$$

Since we connect every document pair with a consistency edge of capacity $\lambda M_{ij}$ and add two terminal edges per document, the number of edges satisfies

$$m = 2k + \Theta(k^2) = \Theta(k^2) = \Theta(n^2), \tag{42}$$

Therefore, the graph is dense.

We compute max flow using the highest-label preflow-push (HLPP) algorithm, which is a push-relabel method that always selects an active vertex of maximum height label and discharges it via a sequence of local push and relabel operations. Unlike augmenting-path methods that repeatedly search for full $s$-$t$ augmenting paths, HLPP only performs local updates on admissible residual arcs, which is particularly suitable for our dense graph where $m = \Theta(n^2)$. For HLPP, a refined amortized analysis based on a potential function yields the worst-case bound (Tunçel, 1994):

$$T_{\text{HLPP}} = O\big(n^2 \sqrt{m}\big). \tag{43}$$

The key idea is to bound the number of non-saturating pushes by splitting them into small and big pushes using a threshold $\kappa$: in each phase, there are at most $O(\kappa n^2)$ small pushes, while the total number of big pushes is bounded by $O(n^2 m/\kappa)$ since each big push decreases the potential by at least $\kappa$. Balancing the two terms by choosing $\kappa = \sqrt{m}$ gives

$$O(\kappa n^2) + O(n^2 m/\kappa) = O\big(n^2 \sqrt{m}\big). \tag{44}$$

In our dense graph $m = \Theta(n^2)$, this further implies

$$T_{\text{HLPP}} = O(n^3). \tag{45}$$

## J. Runtime and Efficiency

We measured average per-query runtime on RealtimeQA. Results for top-$k = 10$ and top-$k = 50$ are shown in Table 8. The findings indicate that RADAR's runtime is primarily dominated by atomic answer generation, while NLI scoring and Min-Cut inference introduce only marginal overhead.

*Table 8.* Runtime and Performance at Top-$k = 10$ and Top-$k = 50$ with attack Pos 1 on RQA using Deepseek.

| Metric | Vanilla RAG | | AstuteRAG | | InstructRAG | | RobustRAG | | ReliabilityRAG | | RADAR (Ours) | |
|---|---|---|---|---|---|---|---|---|---|---|---|---|
| | $k$=10 | $k$=50 | $k$=10 | $k$=50 | $k$=10 | $k$=50 | $k$=10 | $k$=50 | $k$=10 | $k$=50 | $k$=10 | $k$=50 |
| Atomic Gen.(s) | - | - | - | - | - | - | 20 | 86 | 20 | 41 | 20 | 86 |
| NLI(s) | - | - | - | - | - | - | - | - | 0.32 | 0.40 | 0.32 | 0.52 |
| Inference(s) | - | - | - | - | - | - | 0.0365 | 0.0945 | 0.0005 | 0.1072 | 0.0007 | 0.0025 |
| Total(s) | 2 | 2 | 6 | 7 | 3 | 4 | 21 | 94 | 21 | 43 | 21 | 87 |
| Acc. | 25.0 | 35.0 | 25.0 | 21.0 | 23.0 | 33.0 | 64.0 | 69.0 | 69.0 | 67.0 | 69.0 | 72.0 |
| ASR | 74.0 | 65.0 | 1.0 | 5.0 | 68.0 | 46.0 | 7.0 | 9.0 | 15.0 | 18.0 | 11.0 | 5.0 |

*Table 9.* Performance of RADAR and baseline methods with top-$k = 10$ on NQ and TQA.

| Dataset | Pos | Vanilla RAG | | AstuteRAG | | InstructRAG | | RobustRAG | | ReliabilityRAG | | RADAR (Ours) | |
|---|---|---|---|---|---|---|---|---|---|---|---|---|---|
| | | Acc.↑ | ASR.↓ | Acc.↑ | ASR.↓ | Acc.↑ | ASR.↓ | Acc.↑ | ASR.↓ | Acc.↑ | ASR.↓ | Acc.↑ | ASR.↓ |
| **Benign** | | | | | | | | | | | | | |
| NQ | – | 70.2 | – | 27.4 | – | 57.4 | – | 61.6 | – | 67.4 | – | 67.0 | – |
| TQA | – | 76.2 | – | 45.2 | – | 64.2 | – | 60.6 | – | 71.0 | – | 71.6 | – |
| **PIA Attack** | | | | | | | | | | | | | |
| NQ | Pos 1 | 15.0 | 83.6 | 23.0 | 4.4 | 15.8 | 76.4 | 55.6 | 6.8 | **65.2** | 15.0 | 63.4 | 7.8 |
| | Pos 10 | 32.2 | 61.0 | 22.2 | 1.6 | 57.2 | 3.8 | 60.2 | 2.2 | **67.8** | 6.0 | 64.8 | 6.4 |
| TQA | Pos 1 | 13.2 | 88.2 | 35.6 | 5.4 | 19.4 | 73.6 | 59.6 | 21.0 | 60.0 | 35.8 | **61.4** | 32.2 |
| | Pos 10 | 53.6 | 44.0 | 39.6 | 1.8 | 60.0 | 6.4 | 66.4 | 13.2 | **69.6** | 17.4 | **69.6** | 16.0 |
| **Poison Attack** | | | | | | | | | | | | | |
| NQ | Pos 1 | 57.0 | 26.8 | 24.2 | 5.4 | 38.6 | 23.6 | 57.2 | 7.4 | **64.8** | 11.4 | 62.0 | 9.8 |
| | Pos 10 | 66.0 | 13.4 | 31.2 | 1.8 | 49.8 | 5.2 | 60.0 | 2.4 | **65.2** | 5.6 | 64.8 | 6.4 |
| TQA | Pos 1 | 34.8 | 60.2 | 37.0 | 10.4 | 34.0 | 48.4 | 57.4 | 24.8 | 60.4 | 32.4 | **61.6** | 32.0 |
| | Pos 10 | 57.8 | 38.6 | 45.8 | 4.4 | 54.8 | 13.6 | 67.0 | 14.0 | 69.0 | 16.2 | **70.0** | 16.0 |

At top-$k = 10$, RADAR (21 s) matches RobustRAG and ReliabilityRAG in runtime, while achieving the best Acc./ASR under attacks. At top-$k = 50$, RADAR (87 s) is faster than RobustRAG (94 s) but slower than ReliabilityRAG (43 s), as it preserves more comprehensive evidence coverage rather than relying on aggressive subsampling. Vanilla RAG, AstuteRAG, and InstructRAG are more efficient but significantly less robust. Overall, RADAR strikes a favorable robustness–efficiency trade-off, with its additional cost mainly stemming from more complete evidence coverage rather than graph-based reasoning.

## K. Additional Static Results

Under both PIA and Poison attacks, we conduct experiments using Deepseek on the NQ and TQA datasets, as shown in Tables 9– 10. Additionally, under PIA attacks, we evaluate GPT-4o and Grok-4-fast across four datasets (RQA, NQ, TQA, and Bio), as reported in Tables 11–13. Overall, our method demonstrates superior performance compared to the baseline in most cases, achieving higher accuracy and lower ASR, although there are a few scenarios where it performs slightly worse than the baseline.

## L. Experimental Results for Multi-Position Attacks

We conduct multi-position PIA attack experiments on the RQA dataset using Deepseek to examine how the insertion positions within the retrieved list affect model performance.

For top-$k = 10$, we simultaneously attack two retrieval positions. Specifically, we evaluate attacks targeting the early positions (Pos 1 + Pos 2) and the late positions (Pos 9 + Pos 10) to assess how the relative placement of adversarial content influences both Accuracy and ASR. In addition, to better reflect real-world scenarios where the attacker's insertion positions may be uncertain, we sample two positions using a random number generator, resulting in Pos 3 and Pos 5. This randomized setting helps simulate the inherent randomness of practical attacks.

*Table 10.* Performance of RADAR and baseline methods with top-$k = 50$ on NQ and TQA.

| Dataset | Pos | Vanilla RAG | | AstuteRAG | | InstructRAG | | RobustRAG | | ReliabilityRAG | | RADAR (Ours) | |
|---------|-----|------|------|------|------|------|------|------|------|------|------|------|------|
| | | Acc.↑ | ASR.↓ | Acc.↑ | ASR.↓ | Acc.↑ | ASR.↓ | Acc.↑ | ASR.↓ | Acc.↑ | ASR.↓ | Acc.↑ | ASR.↓ |
| **Benign** | | | | | | | | | | | | | |
| NQ | – | 71.6 | – | 32.8 | – | 56.8 | – | 65.6 | – | 68.8 | – | 69.2 | – |
| TQA | – | 76.4 | – | 46.8 | – | 60.8 | – | 68.2 | – | 75.6 | – | 74.0 | – |
| **PIA Attack** | | | | | | | | | | | | | |
| NQ | Pos 1 | 15.2 | 83.4 | 22.0 | 5.2 | 27.8 | 57.0 | 62.0 | 66.0 | 51.2 | 15.4 | **65.0** | 6.6 |
| | Pos 25 | 47.6 | 40.0 | 29.0 | 0.8 | 58.6 | 2.6 | 65.4 | 2.6 | 56.6 | 1.8 | **65.4** | 4.2 |
| | Pos 50 | 41.0 | 48.2 | 27.4 | 1.4 | 59.6 | 4.8 | 66.2 | 2.6 | 66.2 | 3.0 | **66.2** | 4.2 |
| TQA | Pos 1 | 12.4 | 88.8 | 39.0 | 5.6 | 31.0 | 58.0 | 62.0 | 21.6 | 58.8 | 36.0 | **64.6** | 26.2 |
| | Pos 25 | 62.8 | 33.8 | 43.0 | 1.8 | 63.6 | 4.6 | 68.2 | 11.6 | **76.4** | 7.6 | 70.4 | 16.2 |
| | Pos 50 | 62.6 | 30.2 | 43.8 | 2.0 | 63.8 | 5.4 | 68.0 | 11.4 | **76.4** | 5.4 | 70.6 | 16.0 |
| **Poison Attack** | | | | | | | | | | | | | |
| NQ | Pos 1 | 62.2 | 23.0 | 25.2 | 3.2 | 41.2 | 16.8 | 61.8 | 34.0 | 64.4 | 13.4 | **65.4** | 7.0 |
| | Pos 25 | 67.5 | 6.6 | 33.4 | 1.4 | 55.2 | 3.2 | 66.4 | 2.6 | 69.2 | 3.4 | **69.6** | 4.6 |
| | Pos 50 | 67.4 | 5.8 | 31.6 | 1.4 | 56.4 | 4.2 | 66.6 | 2.6 | 68.4 | 3.4 | **69.2** | 4.2 |
| TQA | Pos 1 | 37.4 | 58.4 | 38.6 | 10.6 | 38.2 | 43.0 | 59.4 | 26.2 | 59.8 | 33.4 | **65.0** | 23.2 |
| | Pos 25 | 68.8 | 26.0 | 43.4 | 5.0 | 57.0 | 17.2 | 68.6 | 11.8 | **74.2** | 7.8 | 69.6 | 16.8 |
| | Pos 50 | 73.8 | 15.6 | 43.2 | 4.4 | 58.4 | 15.2 | 68.8 | 11.4 | **75.6** | 4.4 | 70.4 | 15.2 |

*Table 11.* Performance of RADAR and baseline methods under PIA attack on RQA, NQ and TQA using GPT-4o.

| Dataset | Pos | Vanilla RAG | | AstuteRAG | | InstructRAG | | RobustRAG | | ReliabilityRAG | | RADAR (Ours) | |
|---------|-----|------|------|------|------|------|------|------|------|------|------|------|------|
| | | Acc.↑ | ASR.↓ | Acc.↑ | ASR.↓ | Acc.↑ | ASR.↓ | Acc.↑ | ASR.↓ | Acc.↑ | ASR.↓ | Acc.↑ | ASR.↓ |
| **Top-$k = 10$** | | | | | | | | | | | | | |
| RQA | Pos 1 | 57.0 | 32.0 | 27.0 | 47.0 | 21.0 | 62.0 | 69.0 | 16.0 | 68.0 | 24.0 | **69.0** | 16.0 |
| | Pos 10 | 59.0 | 28.0 | 35.0 | 6.0 | 53.0 | 27.0 | 74.0 | 11.0 | 76.0 | 9.0 | **76.0** | 9.0 |
| NQ | Pos 1 | 52.6 | 30.6 | 43.6 | 21.0 | 35.4 | 32.0 | 57.6 | 9.0 | 59.4 | 19.0 | **62.4** | 10.2 |
| | Pos 10 | 47.6 | 30.2 | 43.4 | 3.4 | 51.0 | 17.4 | 62.0 | 4.6 | 65.4 | 8.0 | **66.0** | 7.0 |
| TQA | Pos 1 | 36.8 | 50.6 | 55.2 | 35.2 | 48.8 | 41.4 | 61.2 | 23.4 | 58.0 | 40.6 | **61.4** | 33.4 |
| | Pos 10 | 38.8 | 43.4 | 65.2 | 13.0 | 60.8 | 28.6 | 69.0 | 18.4 | 68.8 | 22.4 | **69.4** | 24.2 |
| **Top-$k = 50$** | | | | | | | | | | | | | |
| RQA | Pos 1 | 53.0 | 39.0 | 32.0 | 41.0 | 27.0 | 55.0 | 69.0 | 15.0 | 67.0 | 23.0 | **74.0** | 13.0 |
| | Pos 25 | 65.0 | 26.0 | 28.0 | 6.0 | 53.0 | 33.0 | 72.0 | 10.0 | 75.0 | 4.0 | **75.0** | 11.0 |
| | Pos 50 | 64.0 | 24.0 | 43.0 | 5.0 | 59.0 | 32.0 | 72.0 | 8.0 | 74.0 | 3.0 | **75.0** | 5.0 |
| NQ | Pos 1 | 50.6 | 34.6 | 46.0 | 16.8 | 34.2 | 35.8 | 57.0 | 8.8 | 58.2 | 19.2 | **64.0** | 7.8 |
| | Pos 25 | 52.6 | 27.0 | 51.6 | 2.2 | 54.2 | 19.8 | 65.4 | 3.2 | **67.8** | 4.2 | 67.6 | 5.2 |
| | Pos 50 | 52.4 | 24.2 | 50.8 | 3.0 | 54.2 | 14.8 | 65.2 | 3.4 | 67.6 | 3.6 | **68.8** | 4.2 |
| TQA | Pos 1 | 32.6 | 50.6 | 59.4 | 30.4 | 48.4 | 38.4 | 63.5 | 28.2 | 60.8 | 35.2 | **63.6** | 32.0 |
| | Pos 25 | 47.2 | 37.8 | 70.0 | 16.4 | 58.2 | 17.1 | 71.0 | 17.0 | **75.8** | 9.0 | 71.2 | 16.2 |
| | Pos 50 | 46.2 | 32.8 | 68.8 | 11.2 | 57.2 | 35.0 | 71.2 | 16.8 | **76.8** | 5.6 | 72.2 | 16.0 |

*Table 12.* Performance under PIA attack with top-$k = 10$ and $k = 50$ on RQA, NQ and TQA using Grok-4-fast.

| Dataset | Pos | Vanilla RAG Acc.↑ | Vanilla RAG ASR.↓ | AstuteRAG Acc.↑ | AstuteRAG ASR.↓ | InstructRAG Acc.↑ | InstructRAG ASR.↓ | RobustRAG Acc.↑ | RobustRAG ASR.↓ | ReliabilityRAG Acc.↑ | ReliabilityRAG ASR.↓ | RADAR (Ours) Acc.↑ | RADAR (Ours) ASR.↓ |
|---|---|---|---|---|---|---|---|---|---|---|---|---|---|
| **Top-$k = 10$** | | | | | | | | | | | | | |
| RQA | Pos 1 | 19.0 | 79.0 | 18.0 | 3.0 | 12.0 | 25.0 | 54.0 | 8.0 | 52.0 | 21.0 | **66.0** | 9.0 |
| | Pos 10 | 37.0 | 57.0 | 18.0 | 3.0 | 47.0 | 5.0 | 56.0 | 4.0 | 60.0 | 10.0 | **69.0** | 5.0 |
| NQ | Pos 1 | 12.2 | 86.0 | 52.2 | 2.6 | 27.2 | 12.2 | 54.4 | 5.2 | 57.0 | 16.8 | **59.4** | 8.2 |
| | Pos 10 | 18.8 | 75.4 | 52.4 | 2.6 | 48.6 | 5.4 | 57.0 | 2.2 | 61.4 | 9.8 | **62.4** | 5.0 |
| TQA | Pos 1 | 11.0 | 89.0 | 65.0 | 3.8 | 51.6 | 25.2 | 66.0 | 16.2 | 62.6 | 30.6 | **66.0** | 18.8 |
| | Pos 10 | 33.4 | 63.8 | 70.8 | 3.6 | 64.4 | 5.2 | 69.0 | 18.4 | 70.2 | 15.2 | **71.0** | 12.8 |
| **Top-$k = 50$** | | | | | | | | | | | | | |
| RQA | Pos 1 | 15.0 | 84.0 | 20.0 | 4.0 | 26.0 | 32.0 | 63.0 | 10.0 | 62.0 | 28.0 | **68.0** | 7.0 |
| | Pos 25 | 45.0 | 42.0 | 20.0 | 3.0 | 52.0 | 3.0 | 72.0 | 4.0 | **76.0** | 8.0 | 75.0 | 2.0 |
| | Pos 50 | 42.0 | 43.0 | 20.0 | 4.0 | 46.0 | 3.0 | 69.0 | 2.0 | **76.0** | 3.0 | **76.0** | 4.0 |
| NQ | Pos 1 | 13.2 | 85.6 | 49.2 | 2.2 | 34.8 | 20.4 | 58.0 | 5.2 | 55.0 | 25.4 | **61.0** | 4.8 |
| | Pos 25 | 31.2 | 60.4 | 50.0 | 3.8 | 51.6 | 3.6 | 61.2 | 2.8 | **62.0** | 3.6 | **62.0** | 4.0 |
| | Pos 50 | 27.4 | 63.2 | 49.8 | 3.0 | 50.4 | 4.0 | 62.0 | 3.2 | **62.0** | 4.0 | **62.0** | 4.0 |
| TQA | Pos 1 | 8.0 | 92.0 | 66.6 | 3.8 | 49.8 | 30.4 | 67.2 | 15.8 | 59.4 | 37.6 | **67.6** | 13.8 |
| | Pos 25 | 41.0 | 55.6 | 66.8 | 3.2 | 66.0 | 4.4 | 70.0 | 10.2 | **74.6** | 8.8 | 70.2 | 10.0 |
| | Pos 50 | 43.2 | 43.6 | 66.6 | 3.0 | 67.6 | 4.4 | 69.8 | 5.6 | **71.8** | 4.2 | **71.8** | 10.0 |

*Table 13.* Performance under PIA attack on Bio using GPT-4o and Grok-4-fast.

| Method | Metric | GPT-4o Top-$k = 10$ Pos 1 | Pos 10 | GPT-4o Top-$k = 50$ Pos 1 | Pos 25 | Pos 50 | Grok-4-fast Top-$k = 10$ Pos 1 | Pos 10 | Grok-4-fast Top-$k = 50$ Pos 1 | Pos 25 | Pos 50 |
|---|---|---|---|---|---|---|---|---|---|---|---|
| Vanilla RAG | Acc.↑ | 36.8 | 8.8 | 39.2 | 28.2 | 8.8 | 57.6 | 9.6 | 54.4 | 38.8 | 13.0 |
| | Rel.↑ | 34.2 | 8.4 | 37.2 | 26.4 | 8.6 | 39.8 | 9.4 | 39.4 | 27.4 | 12.8 |
| | Coh.↑ | 42.0 | 10.2 | 43.6 | 30.6 | 10.6 | 53.4 | 12.4 | 54.6 | 36.4 | 17.0 |
| AstuteRAG | Acc.↑ | 68.2 | 66.8 | 69.8 | 70.0 | 69.8 | 27.4 | 47.0 | 31.8 | 52.6 | 48.8 |
| | Rel.↑ | 69.8 | 62.4 | **67.2** | 67.8 | 66.8 | 22.8 | 24.8 | 24.4 | 27.0 | 26.4 |
| | Coh.↑ | 78.2 | 73.0 | 77.6 | 78.4 | 77.0 | 28.8 | 38.0 | 31.6 | 44.0 | 43.4 |
| InstructRAG | Acc.↑ | 59.4 | 67.6 | 60.8 | 69.6 | 69.0 | 53.8 | 58.2 | 56.0 | 57.4 | 52.6 |
| | Rel.↑ | 47.0 | 66.8 | 51.2 | 64.2 | 64.6 | 29.8 | 30.2 | 32.2 | 31.4 | 26.2 |
| | Coh.↑ | 65.4 | 77.6 | 66.2 | 74.0 | 72.6 | 48.0 | 49.8 | 50.2 | 48.2 | 44.8 |
| RobustRAG | Acc.↑ | 44.6 | 50.8 | 55.2 | 44.6 | 50.8 | 63.6 | 69.6 | 61.0 | 71.2 | **71.4** |
| | Rel.↑ | 40.0 | 45.2 | 58.0 | 40.0 | 45.2 | **59.8** | **64.4** | 53.4 | **56.2** | **55.6** |
| | Coh.↑ | 58.2 | 64.4 | 68.8 | 58.2 | 64.4 | **70.6** | **75.8** | 65.8 | 71.4 | **74.0** |
| ReliabilityRAG | Acc.↑ | 64.8 | 67.6 | 57.6 | 68.4 | 68.0 | 66.0 | 71.8 | 64.0 | 74.0 | 63.2 |
| | Rel.↑ | 64.4 | 66.4 | 58.2 | 66.8 | **67.8** | 43.2 | 47.8 | 38.4 | 48.8 | 40.8 |
| | Coh.↑ | 73.6 | 75.4 | 66.0 | 74.2 | 76.2 | 62.4 | 67.8 | 59.8 | **71.8** | 57.6 |
| RADAR (Ours) | Acc.↑ | **75.2** | **69.2** | **72.0** | **73.4** | **73.6** | **68.0** | **72.8** | **69.6** | **74.2** | 69.8 |
| | Rel.↑ | **68.8** | **68.8** | 65.6 | **70.8** | 67.6 | 55.6 | 55.0 | **54.0** | 53.2 | 50.8 |
| | Coh.↑ | **82.2** | **78.2** | **77.8** | **79.0** | **79.6** | 69.8 | 73.8 | **67.0** | 70.8 | 67.2 |

*Table 14.* Multi-position PIA attack results on RQA using DeepSeek.

| Method | Top-$k = 10$ | | | | | | Top-$k = 50$ | | | | | |
| | Pos 1+2 | | Pos 9+10 | | Pos 4+6 | | Pos 1+2+3 | | Pos 49+50+51 | | Pos 1+4+27 | |
| | Acc.↑ | ASR.↓ | Acc.↑ | ASR.↓ | Acc.↑ | ASR.↓ | Acc.↑ | ASR.↓ | Acc.↑ | ASR.↓ | Acc.↑ | ASR.↓ |
|---|---|---|---|---|---|---|---|---|---|---|---|---|
| Vanilla RAG | 18.0 | 82.0 | 56.0 | 31.0 | 53.0 | 21.0 | 14.0 | 86.0 | 62.0 | 21.0 | 25.0 | 74.0 |
| AstuteRAG | 12.0 | 9.0 | 26.0 | 1.0 | 33.0 | 1.0 | 11.0 | 11.0 | 31.0 | 4.0 | 15.0 | 14.0 |
| InstructRAG | 19.0 | 70.0 | 65.0 | 6.0 | 53.0 | 4.0 | 11.0 | 56.0 | 53.0 | 3.0 | 5.0 | 39.0 |
| RobustRAG | 48.0 | 28.0 | 68.0 | 5.0 | 58.0 | 12.0 | 59.0 | 25.0 | 68.0 | 4.0 | 62.0 | 23.0 |
| ReliabilityRAG | 59.0 | 32.0 | **72.0** | 14.0 | 67.0 | 18.0 | 44.0 | 50.0 | **72.0** | 3.0 | 60.0 | 31.0 |
| RADAR (Ours) | **64.0** | 30.0 | **72.0** | 17.0 | **68.0** | 21.0 | **61.0** | 30.0 | 70.0 | 8.0 | **63.0** | 30.0 |

*Table 15.* Performance under evolving evidence streams with GPT-4o.

| Attack | Pos | Vanilla RAG | | AstuteRAG | | InstructRAG | | RobustRAG | | ReliabilityRAG | | RADAR (Ours) | |
| | | Acc.↑ | ASR.↓ | Acc.↑ | ASR.↓ | Acc.↑ | ASR.↓ | Acc.↑ | ASR.↓ | Acc.↑ | ASR.↓ | Acc.↑ | ASR.↓ |
|---|---|---|---|---|---|---|---|---|---|---|---|---|---|
| Benign | – | 56.05 | – | 58.93 | – | 72.68 | – | 58.46 | – | 58.67 | – | 61.29 | – |
| PIA | Pos 1 | 35.76 | 39.99 | 47.28 | 10.81 | 55.39 | 12.80 | 47.47 | 31.99 | 43.38 | 39.41 | **56.17** | 20.02 |
| | Pos 25 | 45.04 | 25.40 | 50.93 | 10.75 | 53.93 | 32.18 | 53.49 | 26.49 | 56.88 | 8.25 | **60.65** | 12.15 |
| | Pos 50 | 40.12 | 27.90 | 50.99 | 10.68 | 53.42 | 31.35 | 53.29 | 25.14 | 58.09 | 4.54 | **61.23** | 10.87 |

*Table 16.* Performance under evolving evidence streams with Grok-4-fast.

| Attack | Pos | Vanilla RAG | | AstuteRAG | | InstructRAG | | RobustRAG | | ReliabilityRAG | | RADAR (Ours) | |
| | | Acc.↑ | ASR.↓ | Acc.↑ | ASR.↓ | Acc.↑ | ASR.↓ | Acc.↑ | ASR.↓ | Acc.↑ | ASR.↓ | Acc.↑ | ASR.↓ |
|---|---|---|---|---|---|---|---|---|---|---|---|---|---|
| Benign | – | 44.21 | – | 25.27 | – | 46.83 | – | 50.03 | – | 52.72 | – | 56.68 | – |
| PIA | Pos 1 | 5.89 | 75.82 | 25.72 | 2.94 | 4.99 | 27.90 | 44.84 | 38.96 | 41.01 | 43.19 | **53.16** | 20.79 |
| | Pos 25 | 17.08 | 53.04 | 26.10 | 2.82 | 44.79 | 9.40 | 47.98 | 30.13 | 50.93 | 10.36 | **57.58** | 10.23 |
| | Pos 50 | 13.63 | 53.55 | 25.72 | 2.75 | 39.99 | 10.17 | 47.92 | 29.87 | 54.89 | 7.17 | **57.70** | 10.04 |

*Table 17.* Static vs. Dynamic RADAR under the dynamic setting using Deepseek.

| Attack | Pos | RADAR (Static) | | RADAR (Dynamic) | |
| | | Acc.↑ | ASR.↓ | Acc.↑ | ASR.↓ |
|---|---|---|---|---|---|
| Benign | - | 54.13 | - | 74.02 | - |
| PIA | Pos 1 | 48.43 | 20.02 | 63.60 | 17.85 |
| PIA | Pos 25 | 50.86 | 12.98 | 70.12 | 8.94 |
| PIA | Pos 50 | 51.18 | 12.79 | 70.05 | 6.01 |

*Table 18.* Dynamic dataset statistics.

| Statistic | Value (%) |
|---|---|
| Questions with 1 answer change | 77.8 |
| Adjacent-year answer change rate | 68.3 |
| No answer change | 22.2 |
| Exactly 1 change | 26.2 |
| 2 changes | 51.6 |

For top-$k = 50$, we extend the attack to three positions. Following the same protocol, we test attacks at the front of the list (Pos 1 + Pos 2 + Pos 3) and near the end of the list (Pos 48 + Pos 49 + Pos 50). We further include a randomized configuration by sampling three positions with a random number generator, yielding Pos 1, Pos 4, and Pos 27, to evaluate the impact of random multi-point attacks under a larger retrieval budget.

## M. Additional Dynamic Results

We present additional performance results under evolving evidence streams for two different models, GPT-4o which is shown in Table 15 and Grok-4-fast which is shown in Table 16. These results demonstrate that RADAR consistently outperforms baselines in terms of accuracy, particularly in high-ranking evidence positions. Furthermore, RADAR shows superior robustness to evolving attacks, outperforming competing methods in both benign and attack scenarios.

Our dynamic setting does not inject year-specific context into queries, though retrieved documents may contain temporal cues. We compare RADAR (Static) and RADAR (Dynamic) under the same setting to evaluate the static variant and isolate gains from the dynamic extension. As shown in Table 17, the static version degrades more as evidence evolves, while the dynamic version remains more stable, indicating that improvements mainly come from the dynamic design.

## N. Failure-case analysis

**Failure Case 1:**

Query: "*Who won the FIFA Men's World Cup?*"

- 2015 (ground truth: *Barcelona*): Google returns only one informative document (Pos 1). Attacking Pos 1 causes RADAR to output the poisoned answer.

- 2016 (ground truth: *Real Madrid*): Only one informative document appears at Pos 2. Attacking Pos 1 lets the poisoned document dominate centrality; the new correct evidence is filtered out and the memory node preserves the stale answer.

**Failure Case 2:**

Query: "*Who won the Nobel Peace Prize?*"

- 2021 (ground truth: *Maria Ressa and Dmitry Muratov*): Many documents describe the winners unclearly, causing some atomic answers to be generated with only *Maria Ressa*. This leads to the Maria Ressa cluster dominating centrality, pushing the correct full answer aside and causing RADAR to output the wrong answer.

The essence of both failure cases is that RADAR's consensus mechanism using eigenvector centrality and NLI entailment critically relies on the assumption that benign evidence forms the dominant coherent cluster in the retrieved set. When this assumption breaks because genuinely new correct evidence is too sparse or because real-world reporting noise creates a stronger false majority cluster, the Min-Cut selects the wrong partition and the memory node may further reinforces the error.

## O. Dynamic Dataset Statistics

We present descriptive statistics on answer changes over time in Table 18. This shows that answer volatility is substantial, suggesting the dataset better reflects dynamic retrieval settings rather than a mostly static benchmark.

## P. Examples of Dynamic Dataset

Our Dynamic Dataset contains 500 QA questions, each associated with several different years. For each year, we retrieved the top 50 relevant documents from Google. Using Deepseek, we generated incorrect answers for each year, which are used for PIA Attack and ASR statistics, as well as incorrect contexts, which are used for Poison Attack. Here are examples of our dynamic dataset.

**Examples of Our Dynamic Dataset**

```
{
  "question": "Who is the President of the United States?",
  "yearly_contexts": {
    "2015": {
      "answer": [
        "Barack Obama"
      ],
      "docs": [
        {
          "title": "Timeline of the Barack Obama presidency (2015)",
          "url": "https://en.wikipedia.org/wiki/
              Timeline_of_the_Barack_Obama_presidency_(2015)",
          "snippet": "The following is a timeline of the presidency of Barack Obama,
              from January 1 to December 31, 2015. For his time as president-elect,
              see the presidential ...",
          "content": "Timeline of the Barack Obama presidency (2015) - Wikipedia..."
        },
        {
          "title": "Get Ready: President Obama's 2015 State of the Union Address",
          "url": "https://obamawhitehouse.archives.gov/blog/2015/01/11/get-ready-
              president-obamas-2015-state-union-address",
          "snippet": "On Tuesday, January 20 at 9pm ET, President Obama will deliver
              his sixth State of the Union Address. This year there will be more
              ways than ever to take part ...",
          "content": "Get Ready: President Obama's 2015 State of the Union Address
              ..."
        }
        ...
      ],
      "incorrect_answer": "George W. Bush",
      "incorrect_context": [
        "As of 2015, George W. Bush continues to serve as the 43rd President of the
            United States, having been re-elected to a second term in 2004..."
      ]
    },
    "2018": {
      "answer": [
        "Donald Trump"
      ],
      "docs": [
        {
          "title": "2018 United States elections",
          "url": "https://en.wikipedia.org/wiki/2018_United_States_elections",
          "snippet": "Elections were held in the United States on November 6, 2018.
              These midterm elections occurred during incumbent Republican president
              Donald Trump's first ...",
          "content": "2018 United States elections - Wikipedia..."
        },
        {
          "title": "President Donald J. Trump Proclaims January 16, 2018, as
              Religious Freedom Day",
          "url": "https://trumpwhitehouse.archives.gov/presidential-actions/
              president-donald-j-trump-proclaims-january-16-2018-religious-freedom-
              day/",
          "snippet": "On Religious Freedom Day, we celebrate the many faiths that
              make up our country, and we commemorate the 232nd anniversary of the
              passing of a State law.",
          "content": "President Donald J. Trump Proclaims January 16, 2018, as
              Religious Freedom Day..."
        }
        ...
```

```
      ],
      "incorrect_answer": "Barack Obama",
      "incorrect_context": [
        "As of 2018, Barack Obama continues to serve as the 44th President of the
           United States, having been re-elected for a second term in 2012..."
      ]
    }
    ...
  }
  ...
}
```

