# OpenReview forum: "RADAR: Defending RAG Dynamically against Retrieval Corruption"
_ICML.cc/2026/Conference — ICML 2026 regular_

### Official Review · Reviewer_LS8Y · 2026-03-05

**Soundness:** 3
**Presentation:** 3
**Significance:** 2
**Originality:** 2
**Overall Recommendation:** 3
**Confidence:** 3

**Summary:**

This paper proposes RADAR, a defense for Retrieval-Augmented Generation under retrieval corruption, with a particular focus on dynamic web search where the retrieved corpus evolves over time. The method generates per-document atomic answers, builds an entailment and contradiction graph using NLI signals, and selects a coherent evidence subset via an s-t min-cut formulation. For the dynamic setting, it introduces a lightweight memory node updated with a Bayesian-style rule to balance stability and adaptation.

**Compliance With Llm Reviewing Policy:**

Affirmed.

**Final Justification:**

My final recommendation remains a Weak Reject

**Key Questions For Authors:**

see the weaknesses

**Strengths And Weaknesses:**

Strengths
1、The paper addresses a real deployment gap for RAG in time-evolving retrieval settings, where corruption can persist and shift over time rather than appearing as a one-off artifact. And the method is formulated in a clear, principled way by casting context selection as an energy minimization problem and solving it exactly with a Max-Flow Min-Cut construction, which is more convincing than ad hoc filtering.
2、Generating a separate atomic answer for each retrieved document is a reasonable design choice, which reduces cross-document contamination and makes downstream consistency checks more interpretable. This pipeline is clearly illustrated in Figure 1.
3、The evaluation covers multiple corruption types (prompt injection and poisoning) and multiple task settings (short-form QA and longer-form generation), which helps demonstrate breadth beyond a single benchmark.
Weaknesses
1、The pipeline leans heavily on an NLI model to produce the entailment and contradiction scores that drive the whole graph (as shown in Figure 1, with details in Appendices D/E), but the paper does not really show whether these scores stay dependable when the injected text is adversarially written or stylistically “camouflaged,” which feels like a key missing check for this setting.
2、The post-processing step that filters evidence using the embedding agreement threshold λ (Eq. (10)) is reasonable as a heuristic, but it is not very clear why this particular rule is the right signal for “reliability,” as opposed to mostly favoring whatever looks most similar to the majority.
3、The method could be expensive in practice because it needs pairwise consistency scoring across retrieved items, and the objective explicitly includes pairwise terms (Eq. (3)); the appendix touches on efficiency, but the main paper does not give a clear end-to-end latency or cost picture as k grows, which is important for real web-scale RAG.
4、The paper would benefit from a deeper look at failure cases: since RADAR relies on a consensus structure (eigenvector centrality over entailment) plus NLI-based contradiction edges, I would like to see concrete examples where adversarial pages successfully mimic entailment patterns, or where real temporal changes create contradictions and the method ends up filtering the “new but correct” evidence.
5、Presentation is mostly fine, but a few references look inconsistently formatted and some figures are a bit low-resolution; none of this breaks the paper, it just slightly hurts readability and polish.

---

> ### Author Rebuttal · Authors · 2026-03-31
>
> We thank the reviewer for the careful reading and constructive feedback. We appreciate the recognition of our formulation, atomic-answer design, and broad evaluation. We also agree that the discussion of NLI robustness, the embedding-agreement filter, efficiency, and failure modes should be strengthened. In the revision, we will clarify these points and expand the corresponding analysis.
>
> ### W1: Robustness under Adaptive Attack
>
> To directly assess whether the NLI-based entailment/contradiction signals remain reliable under adversarially written or stylistically camouflaged injected text, we additionally evaluated RADAR under the adaptive attack setting proposed by ReliabilityRAG. Specifically, besides the standard non-adaptive injection, we considered the adaptive attack that induces ambiguous answers (e.g., “A or B” while the correct answer is A) to bypass NLI contradiction checks.
>
> Our results in Table 1 show that RADAR remains **largely stable under this stronger attack**: across both top-*k*=10 and top-*k*=50 settings, the performance under adaptive attack is very close to that under non-adaptive attack, with only marginal differences. This suggests that RADAR remains effective even when the injected text is adversarially crafted to camouflage itself against NLI-based defenses.
>
> **Table 1: Performance under none-adaptive attack and adaptive attack (RQA, Deepseek)**
>
> | Top-k | Position | None-Adaptive | Adaptive |
> | ----- | -------- | ------------- | -------- |
> | 10    | Pos 1    | 69 / 11       | 69 / 10  |
> | 10    | Pos 10   | 75 / 5        | 74 / 6   |
> | 50    | Pos 1    | 72 / 5        | 72 / 8   |
> | 50    | Pos 25   | 76 / 3        | 75 / 5   |
> | 50    | Pos 50   | 76 / 3        | 74 / 5   |
>
> ### W2: The motivation for the post-processing step
>
> The post-processing step is not a standalone reliability criterion, but a **conservative outlier-removal step** after graph inference. In RADAR, reliability is primarily determined by the Min-Cut objective, where unary terms encode retention or filtering preferences and pairwise terms enforce consistency. The cosine-similarity filter is applied only afterward to prune semantically isolated atomic answers with very low agreement to the retained set. Since it operates on atomic answers rather than raw passages, Eq. (10) should be understood as a **lightweight claim-level consistency refinement**, rather than a rule that simply favors the majority or serves as the core reliability criterion.
>
> ### W3: The practical runtime as $k$ grows
>
> We measured the average per-query runtime on RealtimeQA, breaking it down into atomic answer generation, NLI scoring, and Min-Cut inference:
>
> - For top-k=10, total runtime is about 21 s: 20 s for atomic answer generation, 0.32 s for NLI, and 0.0007 s for Min-Cut.
> - For top-k=50, total runtime is about 87 s: 86 s for atomic answer generation, 0.52 s for NLI, and 0.0025 s for Min-Cut.
>
> These measurements show that the pairwise NLI stage remains **sub-second in practice**, and the Min-Cut step is **negligible** even though the graph includes pairwise terms. Thus, the bottleneck is atomic answer generation, not graph optimization. Following Reviewers M4i1 and MZW7, we also added runtime comparisons across methods; please refer to Table 4 in our response to Reviewer M4i1 and Table 3 in our response to Reviewer MZW7.
>
> ### W4: Failure-case analysis
>
> Failure Case 1:
>
> Query: “Who won the FIFA Men’s World Cup?”
>
> - 2015 (ground truth: Barcelona): Google returns only one informative document (Pos 1). Attacking Pos 1 causes RADAR to output the poisoned answer.
> - 2016 (ground truth: Real Madrid): Only one informative document appears at Pos 2. Attacking Pos 1 lets the poisoned document dominate centrality; the new correct evidence is filtered out and the memory node preserves the stale answer.
>
> Failure Case 2:
>
> Query: “Who won the Nobel Peace Prize?” (2021, ground truth: Maria Ressa and Dmitry Muratov).
>
> Many documents describe the winners unclearly, causing some atomic answers to be generated with only Maria Ressa. This leads to the Maria Ressa cluster dominating centrality, pushing the correct full answer aside and causing RADAR to output the wrong answer.
>
> The essence of both failure cases is that RADAR’s consensus mechanism using eigenvector centrality and NLI entailment critically relies on the assumption that **benign evidence forms the dominant coherent cluster** in the retrieved set. When this assumption breaks because genuinely new correct evidence is too sparse or because real-world reporting noise creates a stronger false majority cluster, the Min-Cut selects the wrong partition and the memory node may further reinforces the error. We will add a “Failure Cases and Limitations” subsection in the revision.
>
> ### W5: Presentation quality
>
> In the revision, we will carefully clean up the reference formatting and improve figure quality to enhance readability.

---

> > ### Author Rebuttal · Reviewer_LS8Y · 2026-04-03
> >
> > I appreciate the rebuttal. However, my concerns remain unresolved, particularly regarding the second weakness point (i.e., evidence filtering). Therefore, I will maintain my original score.

---

> > > ### Author Response · Authors · 2026-04-03
> > >
> > > Thank you for the follow-up. We appreciate the opportunity to further clarify this point. We hope the following statements will address your concerns and motivate you to adjust your scores appropriately. Furthermore, in the revision, we will make the role of Eq. (10) much more explicit.
> > >
> > > Our intention is not to claim that embedding agreement alone identifies truth. Rather, the post-processing step is a **conservative outlier-removal mechanism** applied only **after the main graph inference** has already selected a candidate reliable subset. In RADAR, the primary notion of reliability is determined by the Min-Cut objective, where unary terms encode retention/filtering preferences and pairwise terms enforce consistency among claims. The role of Eq. (10) is therefore much narrower: it only checks whether an atomic answer that has **already survived graph inference** is still **semantically isolated** relative to the retained set.
> > >
> > > We also do not deny that this step implicitly assumes that, among atomic answers that have already passed the main defense, the answer supported by the dominant consensus cluster is more likely to be correct than a semantically isolated one. In fact, for web retrieval, this is often a **reasonable assumption**: if the majority of retrieved evidence already supports an incorrect claim, then even a human user querying the same evidence pool may also be misled. In that sense, Eq. (10) is intentionally modest: it does not attempt to overturn the consensus formed by the selected evidence, but only removes items that look inconsistent with that consensus at the claim level.
> > >
> > > Importantly, this step is applied after Min-Cut, not before. By the time Eq. (10) is used, the retained subset is already expected to form a **coherent cluster** under the stronger graph-based criterion. Therefore, the post-processing is only relevant in a **relatively small number of residual cases** where the graph inference still keeps a few borderline outliers. A typical example is when two numerical atomic answers such as “approximately 200,000” and “183,000” receive relatively high NLI compatibility scores, so both are retained by Min-Cut even though one is semantically less aligned with the main cluster. In such cases, Eq. (10) acts as a lightweight refinement step to prune these residual outliers, rather than as the core reliability signal.
> > >
> > > We will revise the paper to make two points explicit:
> > >
> > > (1) Eq. (10) is not a standalone reliability criterion and should not be read as “majority similarity = truth”;
> > >
> > > (2) its practical purpose is only to remove rare, semantically isolated atomic answers that remain after the main graph-based selection, especially in cases where NLI-based pairwise scores are overly permissive for near-matching claims such as numerically close answers.
> > >
> > > We sincerely appreciate your constructive feedback, which has significantly helped improve the quality and clarity of the paper.

---

### Official Review · Reviewer_MZW7 · 2026-03-07

**Soundness:** 3
**Presentation:** 3
**Significance:** 2
**Originality:** 2
**Overall Recommendation:** 4
**Confidence:** 4

**Summary:**

This paper studies how to defend dynamic RAG systems from retrieval corruption attacks, where the retrieved evidence changes over time and may include malicious or poisoned content. The authors propose RADAR, a method that turns each retrieved document into a short atomic answer, builds a support/conflict graph using NLI, and uses Min-Cut to keep a reliable set of evidence; for dynamic settings, it also adds a Bayesian memory node to decide when to trust the previous answer and when to update it with new evidence. Overall, the paper’s main contribution is a more structured and memory-efficient defense framework for dynamic RAG, together with experiments on a dynamic benchmark showing improved robustness and much lower storage cost.

**Compliance With Llm Reviewing Policy:**

Affirmed.

**Final Justification:**

The authors’ rebuttal satisfactorily addressed my main concerns through clearer theoretical positioning, additional ablations, and stronger empirical clarification on efficiency and dynamic memory updating. I find the response sufficient overall and therefore raise my score to 4.

**Key Questions For Authors:**

1. What exactly do you mean by “provably robust”?  From the current paper, I can clearly see a proof of exact optimization for the designed objective, but I do not see a formal robustness bound against attacks. Could the authors clarify this point and make the claim more precise?

2. How much does performance depend on atomic answer generation and NLI quality? Please provide stronger ablations to show whether the gains mainly come from the graph formulation itself, or from the upstream atomic-answer / NLI modules. It would be especially helpful to compare different NLI models and different atomic-answer settings.

3. How reliable is the Bayesian memory update in practice? Some likelihood assumptions and approximations are used in the dynamic update. Have the authors validated these assumptions empirically, or compared them with other memory update strategies?

**Limitations:**

No, although the paper includes an impact statement, the discussion of limitations, failure modes, benchmark external validity, and possible misuse remains too limited.

**Strengths And Weaknesses:**

***Strengths***
1. **Important and timely problem.** Dynamic RAG / web-search RAG is closer to real-world deployment than static RAG, so studying its security is meaningful and practical.
2. **Clear and well-structured method.** RADAR is more principled than simple prompt-based self-checking or document-by-document filtering. It combines document reliability and cross-document consistency in one graph optimization framework.
3. **Strong empirical results, especially for dynamic attacks and memory efficiency.** The experiments suggest that RADAR achieves a good accuracy/ASR trade-off in dynamic attack settings, while using only about 1 KB of historical state instead of roughly 30 MB of stored past documents.

***Weaknesses***
1. **The “provably robust” claim seems too strong.**  What the paper clearly shows is that Min-Cut can exactly optimize the proposed energy function. But this is not the same as a formal robustness guarantee against retrieval corruption attacks. Right now, the paper more strongly supports **exact optimization**, not full adversarial robustness.
2. **The method depends heavily on atomic answers and NLI quality.**  If the atomic answers are poor, or if the NLI model makes mistakes on fine-grained entailment/contradiction judgments, then the graph edge weights may be wrong, which can hurt the final evidence selection.
3. **The storage advantage is clear, but the full computation cost is less clear.** The paper emphasizes the 30 MB vs 1 KB memory benefit, which is important. However, RADAR still needs atomic answer generation, pairwise NLI scoring, graph construction, and Min-Cut inference, so the online latency and deployment cost need more discussion.

---

> ### Author Rebuttal · Authors · 2026-03-31
>
> We thank the reviewer for the constructive feedback and for recognizing the importance of dynamic RAG security and the principled design of RADAR. We agree that the theoretical claim and the analysis of upstream dependence, efficiency, and memory updating need further clarification, and we will address these in the final revision.
>
> ### W1 & Q1: The “provably robust” claim
>
> We agree that “provably robust” is imprecise and may be misread as certified robustness. What we prove in Appendix B is that $E(y)$ is submodular, so our s-t Min-Cut yields the global optimum of MAP labeling under the designed MRF, providing **an exact optimization guarantee** for our framework. This is analogous to the “provably robust” of ReliabilityRAG under its assumptions. Since exact Min-Cut inference avoids approximation gaps and local optima, it is especially valuable in dynamic settings where errors can accumulate. In the revision, we will replace “provably robust” with “robust” to distinguish our exact optimization guarantee from certified robustness.
>
> ### W2 & Q2: Gains from Graph Formulation vs. Upstream Modules
>
> We ablated different NLI models (DeBERTa-v3, BART, and ModernBERT) under PIA on RealTimeQA in Table 1-2 and observed **only minor changes** in Acc. and ASR, indicating **low sensitivity** to the NLI choice. We further perturb NLI outputs by randomly replacing them with probabilities of 0.1, 0.3, and 0.5 in Table 3, and observe **only slight accuracy drops**, reinforcing this **low sensitivity**.
>
> For atomic answers, both the main paper and appendix already include results with different LLMs (DeepSeek, GPT-4o, and Grok-4-fast). Moreover, strong and defense-oriented baselines such as RobustRAG and ReliabilityRAG also rely on atomic answers, with ReliabilityRAG further using NLI. Under identical LLM and NLI settings, RADAR still **outperforms both baselines**, suggesting that the main gains come from our **graph-based formulation and exact Min-Cut optimization** rather than the upstream modules. We will clarify this in the revision.
>
> **Table 1: Performance on different NLI (Top-k = 10, PIA attack, RQA, Deepseek, Acc./ASR)**
>
> | NLI        | Pos 1   | Pos 10 |
> | ---------- | ------- | ------ |
> | DeBERTa-v3 | 69 / 11 | 75 / 5 |
> | BART       | 69 / 11 | 75 / 6 |
> | ModernBERT | 69 / 12 | 75 / 6 |
>
> **Table 2: Performance on different NLI (Top-k = 50, PIA attack, RQA, Deepseek, Acc./ASR)**
>
> | NLI        | Pos 1  | Pos 25 | Pos 50 |
> | ---------- | ------ | ------ | ------ |
> | DeBERTa-v3 | 72 / 5 | 76 / 3 | 76 / 3 |
> | BART       | 72 / 7 | 76 / 4 | 76 / 4 |
> | ModernBERT | 72 / 7 | 76 / 4 | 76 / 4 |
>
> **Table 3: Performance under random perturbations of NLI (Top-k = 10, PIA attack, RQA, Deepseek, Acc./ASR)**
>
> |Perturbation Rate|Pos 1|Pos 10|
> |----------|------|------|
> |0|69 / 11|75 / 5|
> |0.1|67 / 10|74 / 6|
> |0.3|68 / 12|73 / 6|
> |0.5|64 / 18|78 / 7|
>
> ### W3: Runtime and efficiency
>
> We measured average per-query runtime on RealTimeQA and decomposed it into atomic answer generation, NLI scoring, and Min-Cut inference. Top-k=10 results are shown in Table 4 of our response to Reviewer M4i1, and top-k=50 results in Table 4.
>
> These results show that atomic answer generation dominates the cost, while NLI and Min-Cut add negligible overhead. At top-k=10, RADAR matches ReliabilityRAG and RobustRAG in latency (21 s) while achieving better robustness. At top-k=50, RADAR (87 s) is faster than RobustRAG (94 s) but slower than ReliabilityRAG (43 s) because it preserves fuller evidence coverage instead of aggressive subsampling. Non-defense baselines are much faster but incur substantially higher ASR under attack. We will add this table and clarify the **robustness-latency-storage trade-off** in the revision.
>
> **Table 4: Runtime and Performance at Top-k = 50 (RQA, Deepseek, Pos 1)**
>
> |Metric|Vanilla RAG|AstuteRAG|InstructRAG|RobustRAG|ReliabilityRAG|RADAR (Ours)|
> |---------------|------------|----------|-------------|-----------|--------------|------|
> |Atomic Gen.(s)|-|-|-|86|41|86|
> |NLI(s)|-|-|-|-|0.40|0.52|
> |Inference(s)|-|-|-|0.0945|0.1072|0.0025|
> |Total(s)|2|7|4|94|43|87|
> |Acc.|35|21|33|69|67|**72**|
> |ASR|65|5|46|9|18|5|
>
> ### Q3: The reliability of Bayesian memory update
>
> We empirically evaluated the Bayesian memory update by comparing RADAR with alternative memory update strategies, including cumulative full-history storage and carry-forward of previous atomic answers; please see Table 3 in our paper and Table 3 in our response to Reviewer M4i1, respectively. RADAR performs best overall, suggesting that RADAR remains effective in practice despite the simplifying likelihood assumptions used in the Bayesian update. This indicates that the main benefit comes from the **dynamic graph mechanism** rather than the exact correctness of the Bayesian assumption. We will clarify this in the revision.

---

> > ### Author Rebuttal · Reviewer_MZW7 · 2026-04-01
> >
> > Thank you for the response. I appreciate the additional clarification and ablation studies, and I'm raising my score to 4.

---

### Official Review · Reviewer_tMqE · 2026-03-11

**Soundness:** 3
**Presentation:** 3
**Significance:** 3
**Originality:** 3
**Overall Recommendation:** 5
**Confidence:** 4

**Summary:**

The paper introduces RADAR, a defense method that protects dynamic RAG systems from corpus poisoning and prompt injection. It selects reliable context by solving a graph-based energy minimization problem using an s-t Min-Cut algorithm. To handle changing information, RADAR employs a Bayesian memory node that recursively updates a "belief state." By weighing past conclusions against new evidence rather than archiving raw documents, RADAR efficiently balances robust defense with the ability to adapt to legitimate knowledge shifts.

**Compliance With Llm Reviewing Policy:**

Affirmed.

**Key Questions For Authors:**

NA

**Strengths And Weaknesses:**

I think the paper is generally technically sound, and the problem it works on is very important and interesting. The idea is easy to understand and novel, as far as I know. The results show promising performance compared to other baseline methods. I think it is an important further step toward improving the robustness of RAG.

Weakness:

I did not find any strong weaknesses in this paper. My major concern is that the paper does not discuss or evaluate adaptive attacks. If the attackers know the given defense, what can they do to improve the attack success rate?

Minor:
There are two reference for Asai, A., Wu, Z., Wang, Y., Sil, A., and Hajishirzi, H. SelfRAG: Learning to retrieve, generate, and critique through
self-reflection. In Proc. of ICLR, 2024a.

---

> ### Author Rebuttal · Authors · 2026-03-31
>
> We thank the reviewer for raising this important point. Our current evaluation follows the **black-box threat model** as described in the paper, where the attacker can inject malicious content into the external corpus but does not have access to the internals of the retriever, generator or the defense mechanism itself. Under a more challenging adaptive setting where the attacker is aware of RADAR, the attacker could improve the attack success rate (ASR) by:
>
> (i) poisoning a larger number of retrieved documents to **weaken the majority of clean evidence**,
>
> (ii) optimizing malicious documents to increase the likelihood of them being **ranked higher**, which can be done by exploiting common retrieval heuristics or using surrogate retrievers,
>
> and (iii) crafting multiple mutually supportive poisoned documents to **simulate semantic consensus**, thereby interfering with RADAR’s centrality- and consistency-based selection process.
>
> We agree that this is an important direction for future research. Notably, our appendix already includes **multi-position attacks**, which partially approximate stronger adaptive attackers, and our experiments show that RADAR maintains relatively **robust performance**, although some degradation is expected under such conditions.
>
> We also thank the reviewer for pointing out the duplicate Self-RAG reference. We will correct this in the final version. Please provide additional comments or feedback. Your insights would be greatly appreciated.

---

> > ### Author Rebuttal · Reviewer_tMqE · 2026-04-04
> >
> > Maintaining my positive score.

---

### Official Review · Reviewer_M4i1 · 2026-03-12

**Soundness:** 2
**Presentation:** 3
**Significance:** 3
**Originality:** 3
**Overall Recommendation:** 4
**Confidence:** 4

**Summary:**

This paper proposes a robust RAG framework for dynamic settings, where documents are retrieved over time and some may be maliciously corrupted. The authors first introduce a novel min-cut-based method that identifies reliable evidence using pairwise entailment and conflict relations among retrieved documents. They then extend this approach to the dynamic setting by incorporating a memory node that summarizes past evidence over time. Across a range of experiments, the method outperforms existing baselines under retrieval-corruption attacks on a newly introduced dynamic RAG benchmark.

**Compliance With Llm Reviewing Policy:**

Affirmed.

**Final Justification:**

During the rebuttal, the authors provide a clearer description of the data and revise the baselines in the dynamic setup, showing improved performance over them, although the improvements appear to be incremental. I find the idea of using a min-cut formulation to be quite interesting, and I lean toward accepting the paper.

**Key Questions For Authors:**

1. Could the authors include more realistic baselines for the dynamic setting? For example, the authors could add the history of previous atomic answers to the prompt and then evaluate how it compares with the full dynamic version. This would help provide a more reasonable comparison between existing methods and the proposed approach.

2. Could the authors elaborate more on the runtime and report a comparison across different methods? A clearer efficiency analysis would help better understand the practical trade-offs between the proposed method and existing baselines.

**Limitations:**

yes

**Strengths And Weaknesses:**

## Strengths

1. The min-cut formulation is interesting and appears novel in this context. The paper also leverages both conflict and similarity between document pairs to infer corruption labels, which seems to be a new contribution within this line of work.

2. The proposed method outperforms prior baselines in the static setting in most cases. In particular, it shows noticeable gains in the hard Pos-1 setup, which may suggest that incorporating similarity information is especially helpful in this regime.

## Weaknesses

My main concern is the way the paper evaluates methods in the dynamic setting.

1. I am not fully convinced that the constructed dataset realistically captures dynamic retrieval settings. The paper would benefit from more discussion and descriptive statistics about the dataset itself, such as how frequently answers change over time and how volatile it is. To better highlight the difference between static and dynamic setups, I suggest that the authors run static attacks in both settings, for example by removing year-specific context from the dynamic dataset, and compare the results.

2. In the dynamic setting, the authors extend prior baselines by storing all previously seen documents, since those methods were originally designed for static retrieval. They then highlight storage-cost improvements over these adapted baselines. I find this extension somewhat naive. There are simpler alternatives for carrying forward past information, such as storing the previous answer $a^{t-1}$ in the prompt or using other lightweight summarization strategies. Because of this, I am not convinced that storage cost is a fundamental limitation of the baselines as presented.

3. While the min-cut formulation appears effective in the static setting, I am less certain about the specific modification introduced for the dynamic setting. It would be helpful if the authors reported results for the static version of their method evaluated directly in the dynamic setup. That comparison would make it much clearer how much of the gain comes from the dynamic extension itself.

---

> ### Author Rebuttal · Authors · 2026-03-31
>
> We thank the reviewer for the constructive feedback and for recognizing the novelty and effectiveness of our min-cut formulation and the value of combining conflict and similarity information. We agree that the dynamic setting needs better justification and evaluation and clarify it in the following response.
>
> ### W1.1: Dynamic dataset statistics
>
> We added descriptive statistics on answer changes over time in Table 1. This shows that **answer volatility is substantial**, suggesting the dataset better reflects dynamic retrieval settings rather than a mostly static benchmark.
>
> **Table 1: Dynamic dataset statistics**
>
> | Statistic                           | Value (%) |
> | ----------------------------------- | --------- |
> | Questions with $\ge$1 answer change | 77.8      |
> | Adjacent-year answer change rate    | 68.3      |
> | No answer change                    | 22.2      |
> | Exactly 1 change                    | 26.2      |
> | $\ge$ 2 changes                     | 51.6      |
>
> ### W1.2 & W3: Static vs. Dynamic RADAR under the dynamic setting
>
> Our dynamic setting does not inject year-specific context into queries, though retrieved documents may contain temporal cues. Following the reviewer’s suggestion, we will compare RADAR (Static) and RADAR (Dynamic) under the same setting to evaluate the static variant and isolate gains from the dynamic extension. As shown in Table 2, the static version degrades more as evidence evolves, while the dynamic version remains **more stable**, indicating that **improvements mainly come from the dynamic design**.
>
> **Table 2: Static vs. Dynamic RADAR under the dynamic setting (Acc./ASR)**
>
> | Attack | Position | RADAR (Static) | RADAR (Dynamic)   |
> | ------ | -------- | -------------- | ----------------- |
> | Benign | -        | 54.13 / -      | **74.02** / -     |
> | PIA    | Pos 1    | 48.43 / 20.02  | **63.60** / 17.85 |
> | PIA    | Pos 25   | 50.86 / 12.98  | **70.12** / 8.94  |
> | PIA    | Pos 50   | 51.18 / 12.79  | **70.05** / 6.01  |
>
> ### W2 & Q1: Dynamic baselines and the storage-cost argument
>
> Regarding W2, we agree that storage cost is not a fundamental limitation of baselines. Following Q1, we implement a stronger baseline that appends only previous atomic answers as compact history, rather than the full document pool. As shown in Table 3, RADAR achieves **better performance** than these baselines in the dynamic setting overall. While InstructRAG is strong in the benign setting, its defense effectiveness under attack is limited. RADAR's superior performance is due to its ability to better handle evolving evidence. We will therefore soften the storage-cost claim and state more precisely that even stronger lightweight-history baselines still underperform RADAR in dynamic defense. We will add this experiment in the revision.
>
> **Table 3: Comparison with dynamic baselines using lightweight history (Acc./ASR)**
>
> |Attack|Position|Vanilla RAG|AstuteRAG|InstructRAG|RobustRAG|ReliabilityRAG|RADAR (Ours)|
> |------|--------|-------------|------------|-------------|-------------|--------------|-----------------|
> |Benign|-|70.70/-|64.17/-|**83.10**/-|59.88/-|72.55/-|74.02/-|
> |PIA|Pos 1|37.68/60.72|60.08/7.74|57.38/32.69|53.49/20.15|50.74/43.12|**63.60**/17.85|
> |PIA|Pos 25|36.98/58.35|64.68/8.25|55.34/25.46|59.31/12.92|66.53/10.81|**70.12**/8.94|
> |PIA|Pos 50|14.84/82.47|62.06/9.85|42.22/49.07|59.12/12.80|68.45/6.78|**70.05**/6.01|
>
> ### Q2: Runtime and efficiency.
>
> We measured average per-query runtime on RealtimeQA. Results for top-k=10 are shown in Table 4, and top-k=50 in Table 3 of our response to Reviewer MZW7. The findings indicate that RADAR’s runtime is primarily dominated by atomic answer generation, while NLI scoring and Min-Cut inference introduce **only marginal overhead**.
>
> At top-k=10, RADAR (21 s) matches RobustRAG and ReliabilityRAG in runtime, while achieving the best Acc./ASR under attacks. At top-k=50, RADAR (87 s) is faster than RobustRAG (94 s) but slower than ReliabilityRAG (43 s), as it preserves more comprehensive evidence coverage rather than relying on aggressive subsampling. Vanilla RAG, AstuteRAG, and InstructRAG are more efficient but significantly less robust.
>
> Overall, RADAR strikes **a favorable robustness–efficiency trade-off**, with its additional cost mainly stemming from more complete evidence coverage rather than graph-based reasoning. We will add these results in the revision.
>
> **Table 4: Runtime and Performance at Top-k = 10 (RQA, Deepseek, Pos 1)**
>
> |Metric|Vanilla RAG|AstuteRAG|InstructRAG|RobustRAG|ReliabilityRAG|RADAR (Ours)|
> |---------------|------------|----------|-------------|-----------|--------------|------|
> |Atomic Gen.(s)|-|-|-|20|20|20|
> |NLI(s)|-|-|-|-|0.32|0.32|
> |Inference(s)|-|-|-|0.0365|0.0005|0.0007|
> |Total(s)|2|6|3|21|21|21|
> |Acc.|25|25|23|64|69|69|
> |ASR|74|1|68|7|15|11|

---

> > ### Author Rebuttal · Reviewer_M4i1 · 2026-04-04
> >
> > I appreciate the authors for providing further analysis of the data and conducting additional experiments with the new lightweight version. I will raise my score to 4.

---

### Decision · Program_Chairs · 2026-04-30

**Decision:**

Accept (regular)

**Comment:**

This paper proposes RADAR, a "robust" Retrieval-Augmented Generation (RAG) framework designed for dynamic settings where documents are retrieved over time and some may be maliciously corrupted. The authors introduce a min-cut-based method that identifies reliable evidence by using pairwise entailment and conflict relations among retrieved documents. This approach is then extended to the dynamic setting by incorporating a memory node that summarizes past evidence. The method is evaluated against existing baselines under retrieval-corruption attacks on a new dynamic RAG benchmark that the authors introduce.

Overall, the reviewers found the paper's approach to be promising. Specifically, the min-cut formulation is interesting and appears novel in this context. The way the paper leverages both conflict and similarity between document pairs to infer corruption labels seems to be a new and valuable contribution within this line of work.

However, during the review process, there were several concerns regarding the evaluation in the dynamic setting. The primary concern was whether the constructed dataset realistically captures dynamic retrieval settings, requiring more statistics on how volatile the answers actually are. Furthermore, extending prior baselines by storing all previously seen documents was seen as somewhat naive. Reviewers pointed out that there are simpler alternatives to carry forward past information, such as storing the previous answer in the prompt, making the authors' "storage-cost" improvement claim less convincing.

During the rebuttal, the authors provided a clearer description of the data's volatility and revised the baselines in the dynamic setup (using a compact history of previous atomic answers). The results showed improved performance over these stronger baselines, somewhat addressing the reviewer's main concerns, although the improvements were seen as incremental.


I agree with the reviewer that the min-cut formulation is an interesting idea, but I have two major concerns that must be addressed:

1) The paper claims to introduce a "provably robust framework", but the formulation reads much more like a heuristic method rather than a well-grounded, formal, or mathematically provable defense. Even the attack model is not rigorously established, nor is the method proved to be robust even in an 'ideal' or 'toy' setting. Thus, the claim of "robustness" should not be understood literally, but rather heuristically or informally.

2) I am also not sure why the proposed method's performance is better than Vanilla RAG under the benign (non-attack) setting. In a purely benign environment without malicious documents, it is unclear why applying a filtering heuristic outperforms simply giving the LLM the standard retrieved context.

Given this, my overall recommendation is a conditional Weak Accept.  The authors must revise the paper to remove any claims or notions that they have "provability" or a "provably robust" framework. They must explicitly state that their method is just a heuristic that seems to work well under their experimental setting and data choices, but actually has not provable guarantees, or any formal robustness (again even the setting is not well specified). Thus, they must explicitly clarify that this heuristic works under their specific experimental setting, only as far as they are aware of. If these conditions are met and the benign performance is better contextualized, the paper is suitable for a weak accept.